# Profiling chromatin accessibility responses in human neutrophils with sensitive pathogen detection

Nikhil Ram-Mohan[1], Simone A Thair[1], Ulrike M Litzenburger[2], Steven Cogill[1], Nadya Andini[1], Xi Yang[1], Howard Y Chang[2], Samuel Yang[1]

**Sepsis, sequela of bloodstream infections and dysregulated host responses, is a leading cause of death globally. Neutrophils tightly regulate responses to pathogens to prevent organ damage. Profiling early host epigenetic responses in neutrophils may aid in disease recognition. We performed assay for transposase-accessible chromatin (ATAC)-seq of human neutrophils challenged with six toll-like receptor ligands and two organisms; and RNA-seq after *Escherichia coli* exposure for 1 and 4 h along with ATAC-seq. ATAC-seq of neutrophils facilitates detection of pathogen DNA. In addition, despite similarities in genomic distribution of differential chromatin changes across challenges, only a fraction overlaps between the challenges. Ligands depict shared signatures, but majority are unique in position, function, and challenge. Epigenomic changes are plastic, only ~120 are shared by *E. coli* challenges over time, resulting in varied differential genes and associated processes. We identify three classes of gene regulation, chromatin access changes in the promoter; changes in the promoter and distal enhancers; and controlling expression through changes solely in distal enhancers. These and transcription factor footprinting reveal timely and challenge specific mechanisms of transcriptional regulation in neutrophils.**

## Introduction

Sepsis, a life-threatening sequela of bloodstream infections due to dysregulated host response, is the leading cause of death related to infections worldwide with rising incidences. Most common bloodstream infection causing bacteria are *Staphylococcus aureus* and *Escherichia coli* with frequencies of 20.5% and 16%, respectively, in culture-positive patients (1). Time is of the essence in sepsis, as every hour delay in appropriate antibiotic therapy decreases survival by 7.6% (2). Understanding early host–pathogen interplay in sepsis can offer invaluable clinical insights critical to saving lives. Neutrophils are the first responders to infection and have been extensively studied for their role in infection and inflammatory processes, particularly sepsis (3). Neutrophils recognize pathogen associated molecular patterns (PAMPs) via TLRs (4) and danger associated molecular patterns (DAMPs) via receptors such as receptor for advanced glycation of end products (RAGE) for high mobility group box 1 (HMGB1) (5). PAMPs are derived from the cell walls of live or dead pathogenic organisms (exogenous signals), whereas DAMPs are derived from the host (endogenous signals) and each are specifically recognized by different TLRs (6). Both result in inflammatory responses involved in sepsis. Neutrophils responding to PAMPs and DAMPs are capable of unleashing immediate, antimicrobial effector functions, including neutrophil extracellular trap (NET) production, phagocytosis, superoxide production, and release of cytokines for further recruitment of other neutrophils and macrophages in a tightly regulated manner (7, 8). Moreover, studies have described that the neutrophil life span may be extended from 5–8 h in the periphery to days upon interaction with both PAMPs and DAMPs (9, 10). These responses are tightly regulated to avoid collateral damage such as increased vascular permeability and hypotensive shock resulting from release of heparin binding proteins by neutrophil activation via adherence to endothelial cells (11) and lung injury and poor patient outcome because of a cytokine storm resulting from hyperresponsiveness and dysregulation of apoptosis in lung neutrophils (12).

Despite possessing tightly regulated yet diverse functions, neutrophils have been regarded as a terminally differentiated cell type with limited ability to produce transcripts or proteins. The inference from this assumption is that the chromatin structure of a neutrophil is dynamically limited. Specifically in comparison to monocytes, neutrophils were shown to have much lower gene expression and largely repressed chromatin (13 Preprint). However, despite the fact that neutrophils have reduced transcriptional activity overall, they possess a much more dynamic range of transcripts and 5′—C—phosphate—G—3′ (CpG) patterns when compared with other cell types (14). Neutrophils also show heterogeneity in their methylation patterns between individuals (15) and undergo active chromatin remodeling, methylation/acetylation patterns associated with gene transcription, and cytokine production (16, 17, 18, 19). Neutrophils use an inhibitor program to safeguard their

[1]Department of Emergency Medicine, Stanford University School of Medicine, Stanford, CA, USA    [2]Center for Personal Dynamic Regulomes, Stanford University, Stanford, CA, USA

Correspondence: syang5@stanford.edu

epigenome from unregulated activation, thereby protecting the host (20). Epigenetic signatures have also been shown to play a role in the cellular function of septic patients (21). Specifically, bacteria can affect the chromatin structure of host immune cells via histone modifications, DNA methylation, restructuring of CCCTC-binding factor (CTCF) loops, and noncoding RNA (19, 22, 23). Such chromatin changes allow for repositioning of inflammatory genes into a transcriptionally active state, recruitment of cohesion near the enhancer regions, and result in swift transcriptional response in the presence of *E. coli* (19). Even though chromatin remodeling is shown in neutrophils in response to external stimuli, the exact regulation of transcription by changes in chromatin accessibility is not well understood.

Because the epigenome reacts before gene expression, we are interested in profiling chromatin responses in neutrophils to infections for early disease recognition. In the present study, we first explore the relevant chromatin elements involved in TLR-mediated responses to 1 h exposures from various pathogen ligands or whole *S. aureus* and *E. coli* on a genome-wide scale using assay for transposase-accessible chromatin (ATAC)-seq to elucidate differences in the host response. We also assay the temporal fluctuation between 1- and 4-h post *E. coli* exposure in the neutrophil epigenome and resulting transcriptome to better understand the processes and pathways involved in immune response. Neutrophilic chromatin accessibility patterns may reveal what pathogen an individual has encountered and/or how they are responding to the infection. Our analyses reveal chromatin accessibility, enriched motifs, and functional signatures unique to each challenge. We also observe time specific chromatin accessibility changes resulting in transcriptional changes at two time points. Based on the chromatin accessibility patterns, we classify three categories of transcriptional regulation in neutrophils. In addition, because prokaryotes lack chromatin, using ATAC-seq results in increased pathogen to host ratio of DNA, enhancing rare microbial reads. The coupling of host response profiles with microbial reads in a single assay may offer diagnostic advantages while gaining unprecedented insights into early host–pathogen dynamics and neutrophil biology.

# Results

## Neutrophils are activated in response to ligand and whole organism challenges

Purified neutrophils from four female healthy volunteers were challenged with an array of TLR agonists for 1 h, namely, lipotechoic acid (LTA) (TLR2) (24), LPS (TLR4) (5), flagellin (FLAG) (TLR5) (4), resiquimod (R848) (TLR7/8) (4), and β-glucan peptide (BGP) that signals via the dectin-1 receptor (25) for fungal infection and the DAMP HMGB1, a cytokine released in sterile inflammation (such as early traumatic events, thought to signal through RAGE and TLR4) (5, 26) (Fig 1B). Neutrophil activation was confirmed by IL8 and TNFα qRT-PCR (Fig 1C). Because it is important that the ATAC-seq is performed on intact nuclei, SYTOX green assay was performed to estimate the extracellular DNA as an indication of NETs. No NETs were observed in response to any stimuli at the time of ATAC-seq (1 or 4 h of stimulation, Fig 1D)

supporting earlier findings that neutrophil nuclei remain intact 1 h after stimulation with PMA to trigger NETosis (27).

## Pathogen DNA from challenges is enriched in ATAC-seq

Neutrophils possess largely closed chromatin, and prokaryotes lack chromatin. ATAC-seq on neutrophils yields several features that increase sensitivity for pathogen reads. First, negative isolation of neutrophils reduces the number of human cells and potentially captures any circulating or phagocytized pathogens. Second, by surveying only open chromatin, ATAC-seq increases the pathogen to host ratio of DNA in the sample compared with traditional library preparation methods. To demonstrate this, whole blood was challenged with *S. aureus* in incremental colony forming units (CFU) per ml for 1 h and neutrophils were negatively isolated. These were then parallelly subjected to genome-wide sequencing using a standard solid-phase reversible immobilization (SPRI) library preparation and ATAC-seq. As suspected, the relative abundance of *S. aureus* reads obtained by ATAC-seq was higher than the SPRI method, for all concentrations (Fig 1E). In fact, the relative abundance retained by ATAC-seq at $10^3$ CFU/ml is comparable to the $10^5$ CFU/ml SPRI preparation method, which is a marked improvement in sensitivity. Contaminant signals are present, after removing all human reads, given that the neutrophils were only challenged with *S. aureus*, these are likely short, low-complexity reads that that do not map specifically. Despite the contamination, ATAC-seq samples display three times more reads for the pathogen compared with SPRI.

## Differential accessibility of chromatin in the genome is challenge and time specific

Neutrophils have limited accessible chromatin; however, insert size distributions and enrichment at transcription start sites (TSSs) were consistent across samples (Fig S1A and B). Strong correlation of genome-wide peak counts across technical replicates (from $r^2 = 0.70–0.95$) were observed for any given ligand stimulation (Fig S2), whereas lower correlation was observed between donors, suggesting donor specific heterogeneity.

Differential accessibility of chromatin in neutrophils is readily apparent when comparing the accessible chromatin landscape in response to the challenges with that of unstimulated neutrophils using Diffbind with $P < 0.05$ and abs(logFC) ≥ 1 (Tables 1 and S1). Despite the differences in the numbers of differential regions in each challenge, their genomic distribution identified using ChIP-seeker (Fig 2A) is similar. More than 40% of the differential regions in each challenge were found in either distal intergenic or intronic regions.

Landscape of differential chromatin accessibility is challenge dependent. This uniqueness is down to the gene level where associated differential regions between challenges are disparate (Fig 2B). A map of the differential regions around the *TLE3* gene portrays this uniqueness clearly. Overlap analyses of differential regions across the entire genome between challenges showed that the vast majority of these regions did not overlap and were challenge specific (Fig 2C and D and Table 1). There are no overlapping differential regions across all of the challenges. Six of the nine

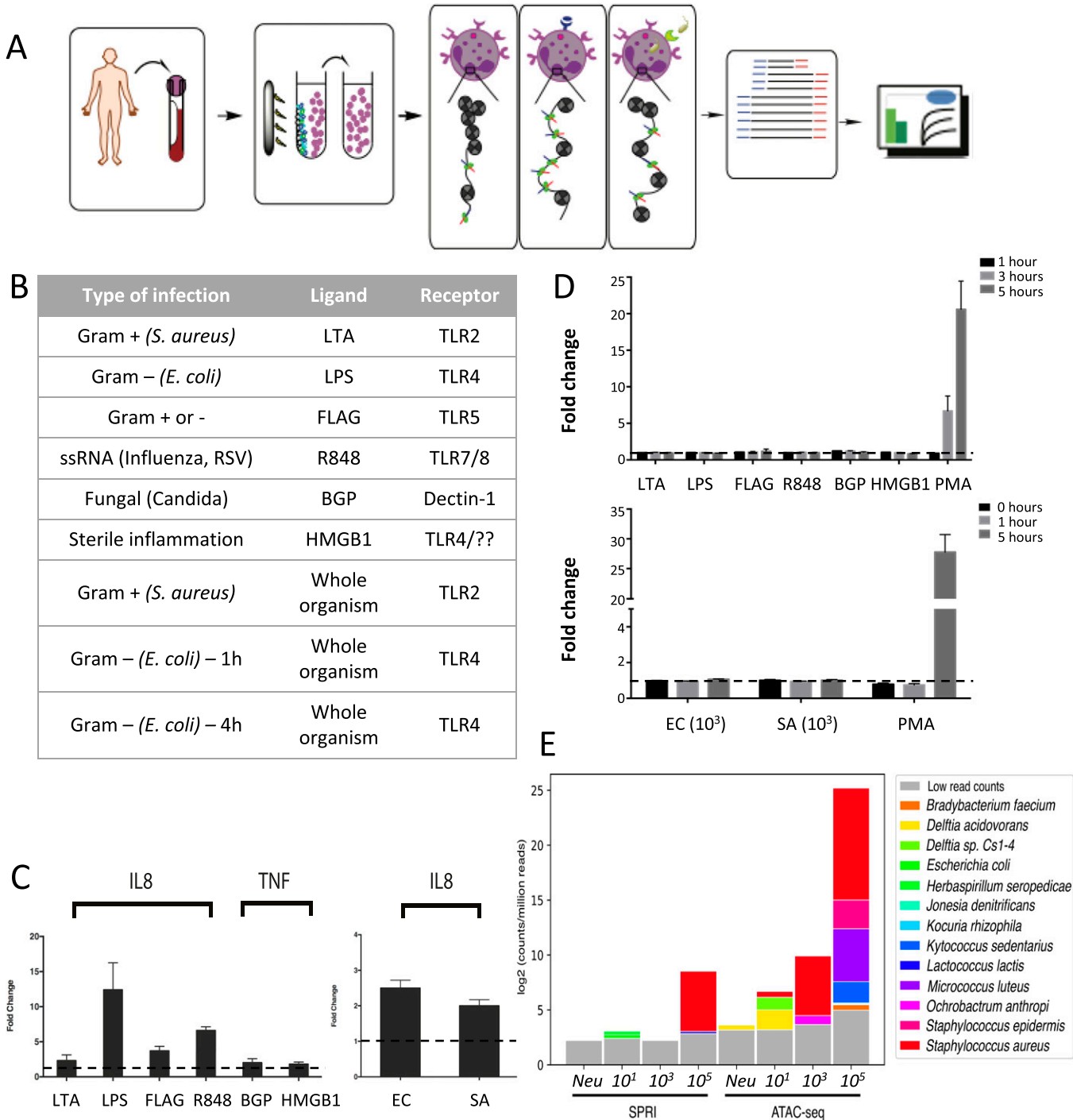

**Figure 1. Neutrophil activation in response to challenges.**
**(A)** Schematic of neutrophil isolation, stimulation, and ATAC-seq. Blood is collected from healthy volunteers in EDTA tubes and unwanted cells are removed using a magnetic bead selection. Tn5 transposase (green ovals) carrying an adaptor payload (red and blue) complementary to next generation flow cells and inserts randomly into regions of open chromatin. Unstimulated and stimulated neutrophils are the sequenced using Illumina technology. **(B)** Table of tested challenges including six ligands, two whole organisms, and one time series. **(C)** Healthy donor neutrophils produce IL8 or TNF in response to ligands or live organism challenge (ligand donors, n = 4; live organism donors, n = 2; mean and SE are represented). **(D)** Healthy volunteer neutrophils do not produce neutrophil extracellular traps via SYTOX green assay in response to pathogen ligands at 1 h or immediately after live organism challenge supporting this time point for ATAC-seq. (PMA is a positive control) (ligand donors n = 4, live organism donors n = 2). **(C, D)** Dashed lines in (C, D) indicate a fold change of 1. **(E)** ATAC-seq is more sensitive to *Staphylococcus aureus* reads than traditional SPRI library preparation. Whole blood was spiked at increasing concentrations with live *S. aureus* and neutrophils (Neu) were negatively isolated. These were prepared for sequencing using traditional DNA extraction and library preparation (SPRI method) compared with neutrophil isolation and ATAC-seq method (n = 2) (Neu: isolated neutrophils, no organisms). Relative abundance plots illustrate that reads align to *S. aureus* (red) and other bacteria; however, the species that would be contamination are still low in relative abundance.

**Table 1.  Number of DRs in response to challenges when compared with the unstimulated controls.**

| Challenge | Number of DRs | Number of unique DRs (%) | Number of DRs associated with genes |
|---|---|---|---|
| LTA | 1,331 | 850 (64) | 1,277 |
| LPS | 1,729 | 1,133 (65) | 1,653 |
| FLAG | 2,963 | 1,967 (66) | 2,857 |
| R848 | 3,105 | 2,135 (69) | 3,034 |
| β-glucan peptide | 2,030 | 1,530 (75) | 1,958 |
| High mobility group box 1 | 2,930 | 2,234 (76) | 2,845 |
| *Staphylococcus aureus* | 2,241 | 2,121 (95) | 1,999 |
| EC-1h (EC1h) | 5,010 | 4,625 (92) | 4,863 |
| EC-4h (EC4h) | 1,688 | 1,452 (86) | 1,633 |

challenges tested were the most to have any overlapping differential regions. A total of five regions were shared by a combination of six challenges, 45 were common between five challenges, 120 between four challenges, 359 between three challenges, and 1,582 regions were common between a combination of two challenges. Interestingly, of the five differential regions common to six challenges, two are common across all the ligand challenges (Fig 2C). There is only one common differential region between the whole organism challenges. However, comparing the signature between the whole organism challenge and their corresponding ligands, a few commonalities exist (Fig 2D). There are three common differential regions between the *S. aureus* and LTA challenges, 118 common between the two *E. coli* time points, EC1h and LPS have 19 in common, and EC4h and LPS have 3 in common despite the different stimulation strategies (neutrophil versus whole blood). Surprisingly, however, there are no differential regions common to the two *E. coli* time points and LPS. On average, ~69.37% of the differential regions from the merged peak sets are unique to the ligand challenges, whereas with the whole organism challenges, ~91% of the regions are unique to the challenge. In addition, comparing EC1h and EC4h differential regions with Hi-C defined interacting regions after 3 h of treatment with *E. coli* show overlap between the ATAC-seq differential regions and Hi-C predicted interacting regions at the whole genome level. 4,506 EC1h differential regions overlap with one end of the Hi-C interactions, whereas 4,498 overlap with the other. Similarly, for EC4h, 1,494 differential regions overlap with one end of the interacting regions and 1,471 overlap with the other.

Unique chromatin accessibility signatures in response to challenges are not limited to specific positions of differential chromatin accessibility but correspond to differences in the regulated functional pathways as well. Differential regions were assigned to genes by using a combination of prediction tools and surveying overlaps with known regulatory regions. This was done rather than assigning regions to genes immediately downstream so as to account for distal regulation as well. Prediction methods included T-gene (28 Preprint) and Genomic Regions Enrichment of Annotations Tool (GREAT) (29). Overlap analyses were performed against the human active enhancer to interpret regulatory variants (HACER) database as well as the predicted regulatory regions identified in primary human cancers (30). Applying this combinatorial method, a

minimum, that is, ~89.2% of the differential regions were associated with genes, whereas, on average, ~95.8% of the differential regions were successfully associated with genes across all challenges (Table 1). In addition, on average, 89.1% of these overlapped previously identified regions with histone marks from the IHEC. GO term and pathway enrichment analyses with the assigned genes and the number of differentially induced and repressed regions associated with each of them revealed dissimilar functional enrichment signatures between challenges (Figs 3A and S3). There are no common enriched pathways with associated differentially accessible regions (DRs) across all challenges, and, interestingly, there are no enriched pathways associated with the LPS and *S. aureus* challenges. Very different pathways are enriched based on the associated induced and repressed DRs for the two *E. coli* challenge time points as well. Repressed DRs are enriched in neutrophil degranulation and signaling by interleukins at EC1h, whereas induced DRs are enriched in programmed cell death, signaling by interleukins, and platelet activation at EC4h. Overall, other than for a few overlapping enriched pathways, each challenge elicited varied functional responses upon stimulating the neutrophils.

Similar to patterns with position and function, there are unique enriched motifs specific to most challenges. Importantly, however, despite large proportions of differential regions being unique to each challenge, these still contain common enriched motifs shared across multiple challenges. For example, three enriched motifs in induced DRs are common to 6/9 challenges and 10 are common to five of the challenges (Fig 3B). In addition, comparing enriched motifs in all the induced and repressed DRs across the challenges, three of the nine challenges showed unique enriched motifs. There were 0 unique motifs in the BGP challenge, 0 in the FLAG challenge, 0 in HMGB1, 0 in LPS, 0 in LTA, 3 in R848, 6 in *S. aureus*, 3 in EC1h, and 0 in EC4h (Tables S2 and S3). Hence, it is important to combine both specific positions as well as the enriched motifs to identify signatures unique to each challenge.

### Transcriptional plasticity of neutrophils in response to *E. coli* challenges

RNA-seq analysis of *E. coli* challenged neutrophils at two time points—1 (EC1h) and 4 h (EC4h)—using edgeR revealed a temporal

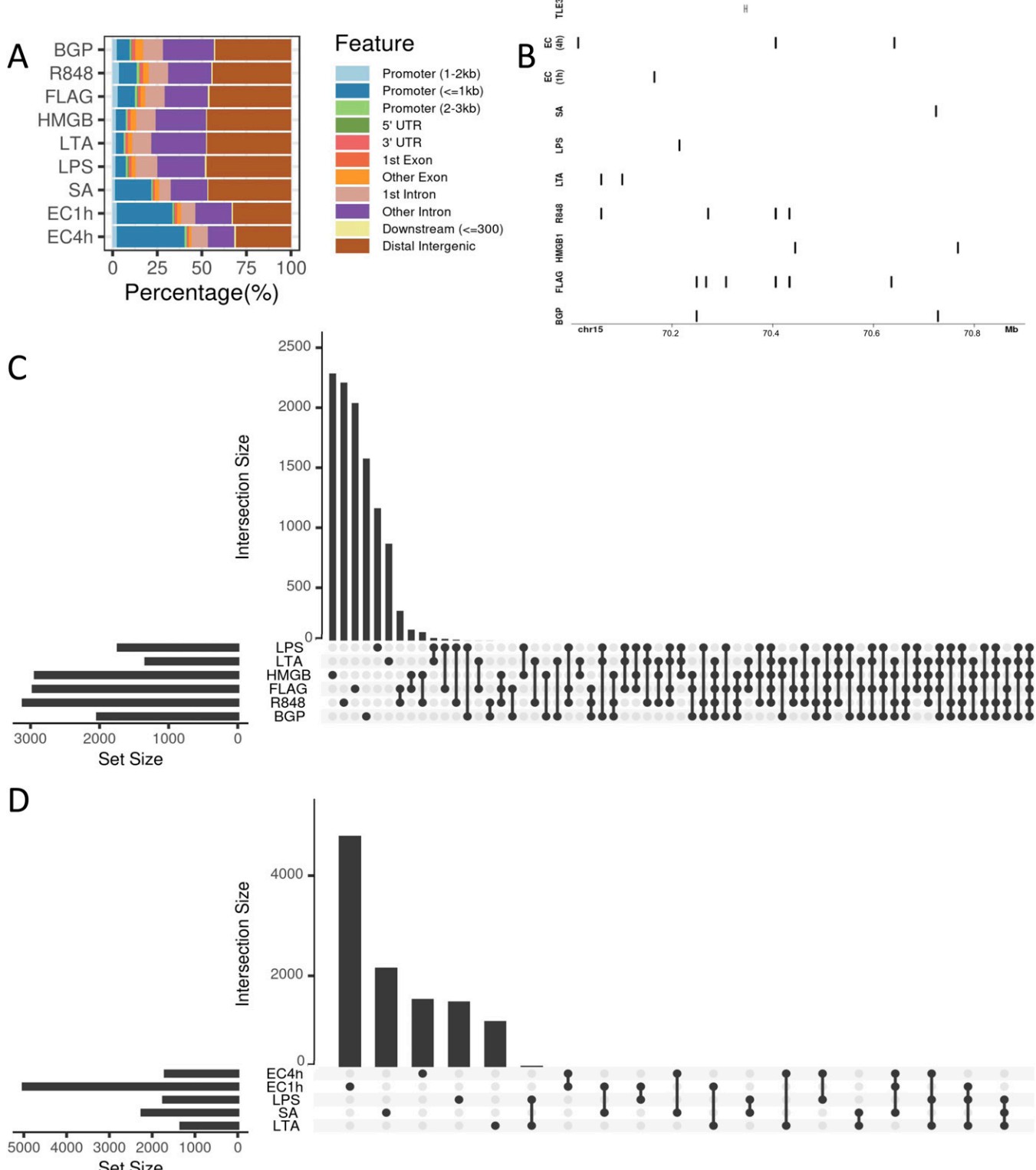

**Figure 2. Comparison of locations of differential regions across challenges.**
**(A)** Whole genome distribution of differential regions within promoters, UTRs, exons, introns, downstream regions, and distal regions as determined using ChIPseeker. Very similar distribution patterns across sample, more than 80% of differential regions are distal. **(B)** Depiction of the unique signatures across the challenges. Differential regions around the *TLE3* gene. From top, the *TLE3* gene, differential regions associated at EC4h, EC1h, *Staphylococcus aureus*, LPS, LTA, R848, high mobility group box 1, FLAG, and β-glucan peptide, respectively. **(C)** Upset plot of the top 100 overlapping differential regions across ligand challenges. Consensus set of regions were generated using Diffbind and the presence absence was visualized using Upset on R. Most differential regions are unique to challenges. **(D)** Upset plot of the top 100 overlapping regions between the whole organism challenges and their corresponding ligands. Minimal overlap between whole organisms and ligands.

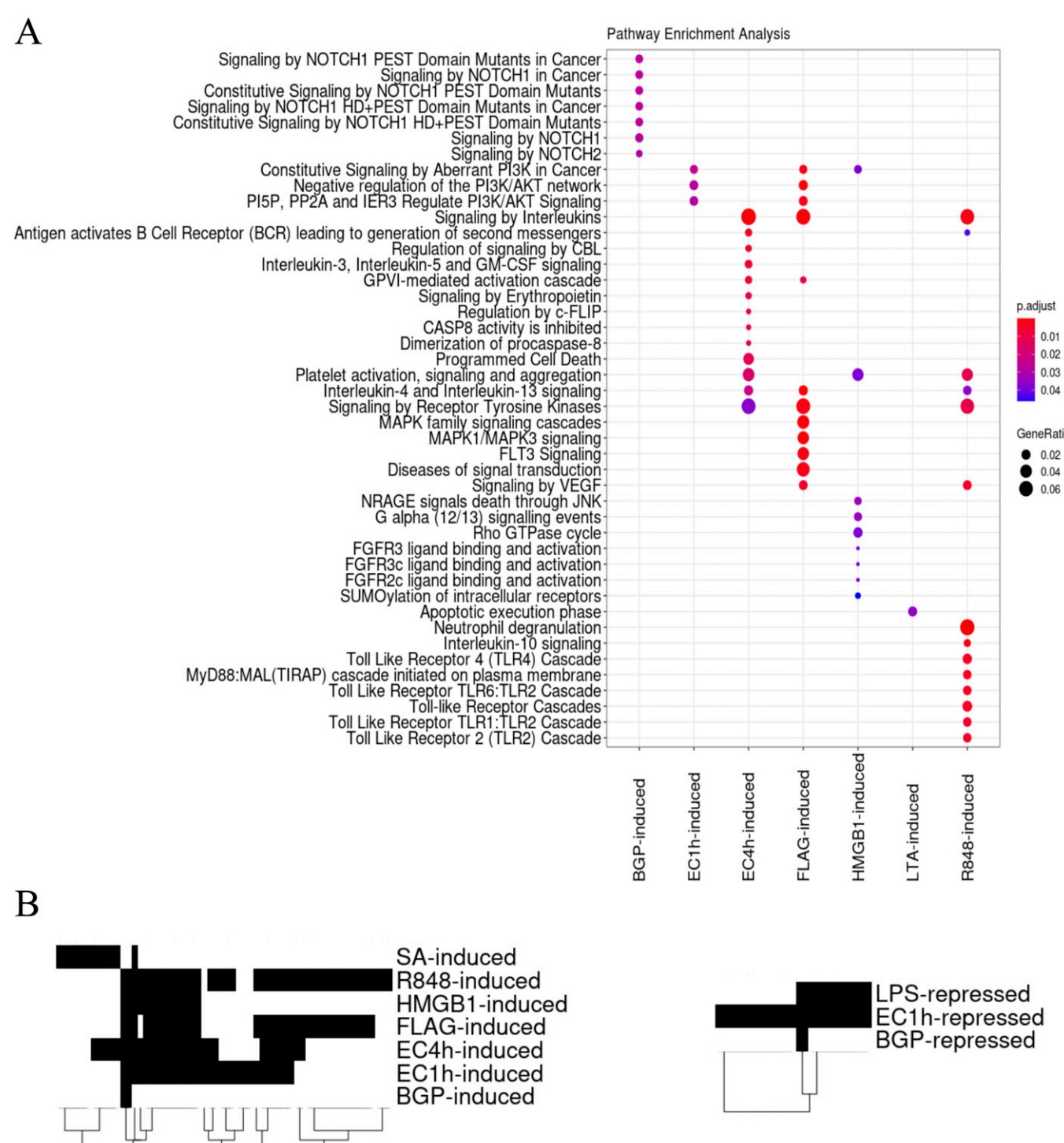

**Figure 3. Comparison of functional profiles of differential regions.**

**(A)** Reactome pathway enrichment analysis of the genes associated with the differential regions in each challenge. Gene assignments to each peak were carried out as described in the methods. Enrichment analysis performed using ChIPseeker and compared using the compareCluster function from clusterProfiler. **(B)** Presence–absence heat map of enriched motifs in the differential regions from each challenge. For induced and repressed regions in each challenge with respect to the untreated controls, enriched motifs were determined using HOMER and filtering for a *P*-value of $P < 10^{-10}$. We compared enriched motifs in induced (left) and repressed (right) differential regions across challenges. Although there exist shared enriched motifs, there are motifs unique to three of the nine challenges.

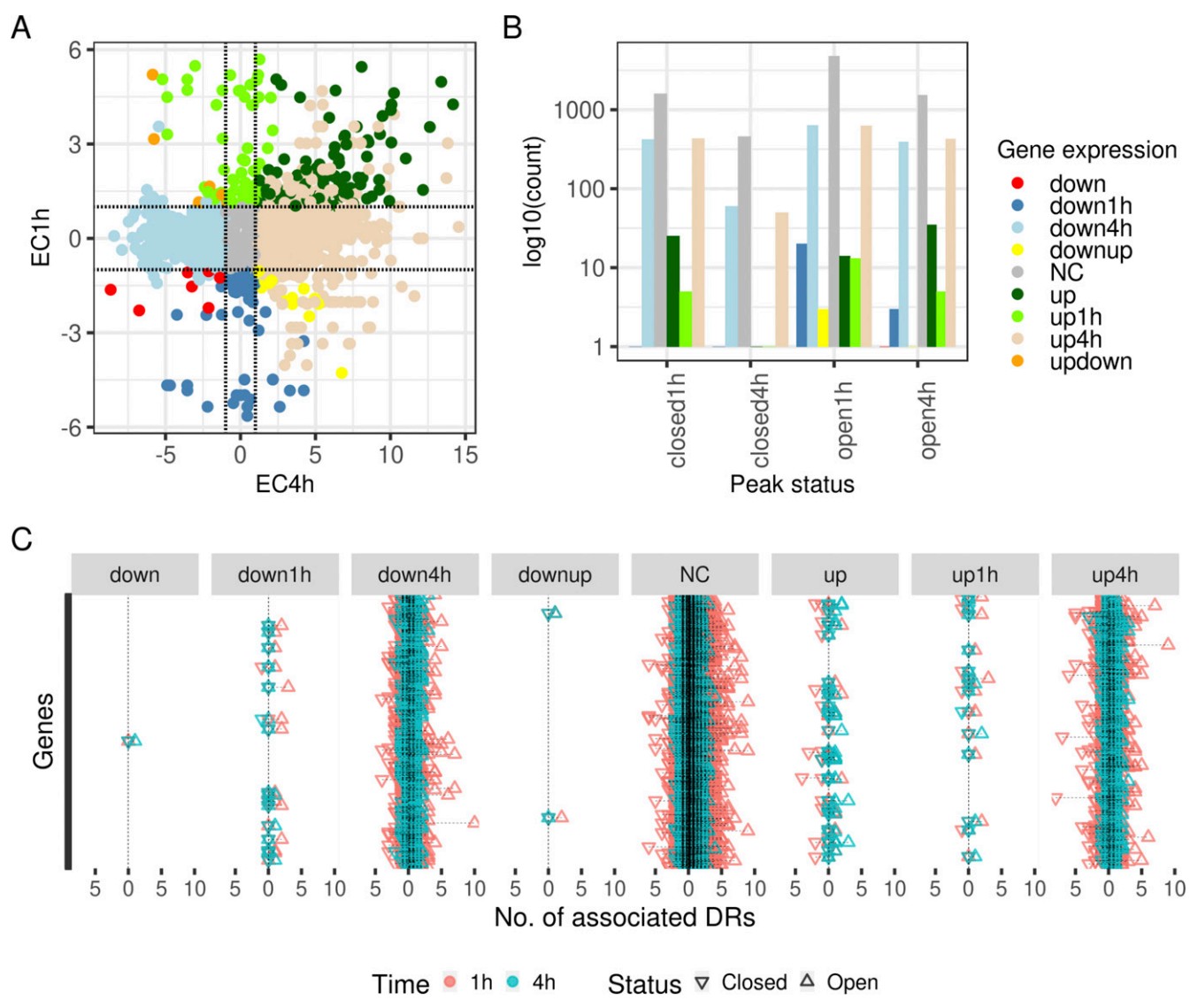

**Figure 4. Paired ATAC-seq and RNA-seq of neutrophils challenged with *Escherichia coli* at 1 and 4 h.**
**(A)** Comparison of expression of genes at the two time points. logFC calculated using edgeR for each gene at the two time points were plotted as points if the *P*-value is <0.05. Points are colored based on the gene expression patterns at the two time points. down: genes down-regulated at both time points; down1h and down4h: genes down-regulated only at 1 and 4 h time points, respectively; downup: genes down-regulated at 1 h and up-regulated at 4 h; NC: genes that are not differentially expressed at either time point; up: genes up-regulated at both time points; up1h and up4h: genes up-regulated only at 1 and 4 h time points, respectively; and updown: genes that are up-regulated at 1 h and down-regulated at 4 h. **(B)** Distribution of open and closed DRs at each time point with respect to the gene expression patterns. Combination of open and closed regions at each time point and each gene expression pattern except for genes that are down-regulated at 1 and 4 h. **(C)** Counts of open and closed DRs associated with each gene at each time point. Typically, more associated DRs for each gene at 1 h than at 4 h. In addition, combination of multiple open and closed DRs associated with most genes.

pattern in gene expression suggesting plasticity in neutrophil transcription with strong correlation between replicates ($r^2$ ranging from 0.92 to 0.99). Although most genes remain unchanged in expression in the presence of *E. coli*, there are differences in the number of differentially expressed genes and the magnitude of change between the two time points (Fig 4A). At EC1h, there are 66 up- and 55 down-regulated genes, and at EC4h, there are 2,554 up- and 2,656 down-regulated genes. 93 genes are up- and 10 genes are down-regulated at EC1h and EC4h. Interestingly, there are a few genes that are either up-regulated at 1 h and down-regulated at 4 h

or vice versa. In this category, 7 genes were up-regulated at 1 h and down-regulated at 4, and 16 were down-regulated at 1 h and up-regulated at 4 h (Table S4).

Plasticity is also reflected in the biological processes enriched at the two time points (Fig S4). GO term enrichment of genes grouped in above mentioned categories portrays a transcriptional landscape in neutrophils that are responsive to external stimuli. For example, immunoglobulin production is transiently up-regulated at 1 h, suggesting immediate response to the challenge. Transport and metabolism are predominant in the down-regulated genes at 1 h.

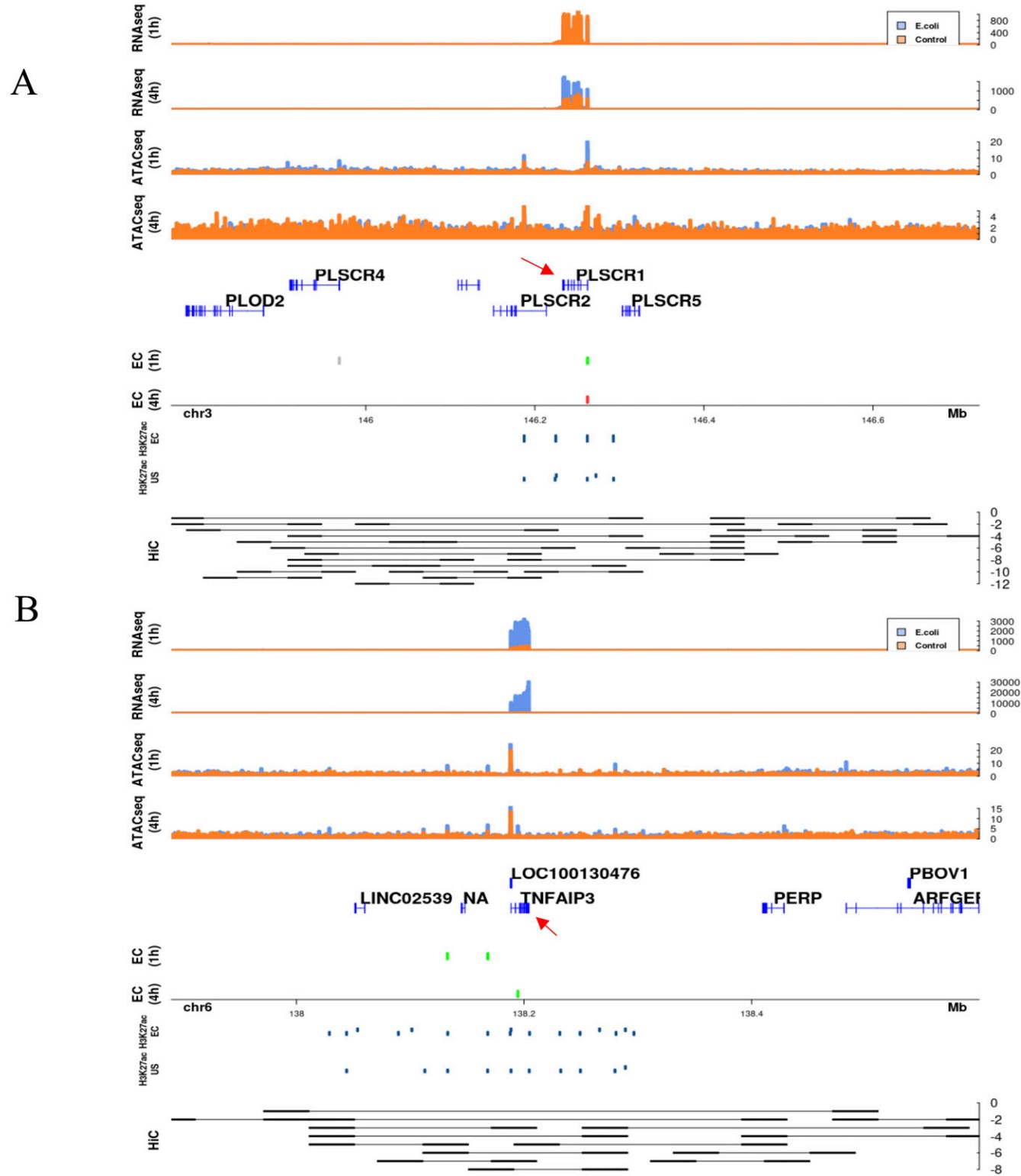

**Figure 5. Mechanisms of control of transcription by accessible chromatin in neutrophils.**
**(A)** Differential regions only in the promoter. Promoter is defined as the regions ±2.5 kb around the transcription start site for each gene. Example for this category is the *PLSCR1* gene. **(B)** Differential regions only in distal regions, promoter is primed for expression. Example for this category is the *TNFAIP3* gene. **(A, B)** In (A, B), from the top, RNA-seq coverage at 1 h for *Escherichia coli* and control; RNA-seq coverage at 4 h; ATAC-seq coverage at 1 h; ATAC-seq coverage at 4 h; genes in the region from hg19; open (green) differential regions associated with the gene at 1 h and open or closed differential regions not associated with the gene of interest (gray); differential regions associated at 4 h; location on the chromosome; H3K27ac histone marks in the presence of *E. coli*; and absence of *E. coli* lifted over from earlier study (19); and Hi-C interactions in the presence of *E. coli* lifted over from earlier study (19) that show overlap between ATAC-seq defined differentially accessible chromatin regions and Hi-C interacting regions. Red arrows point to the gene of interest.

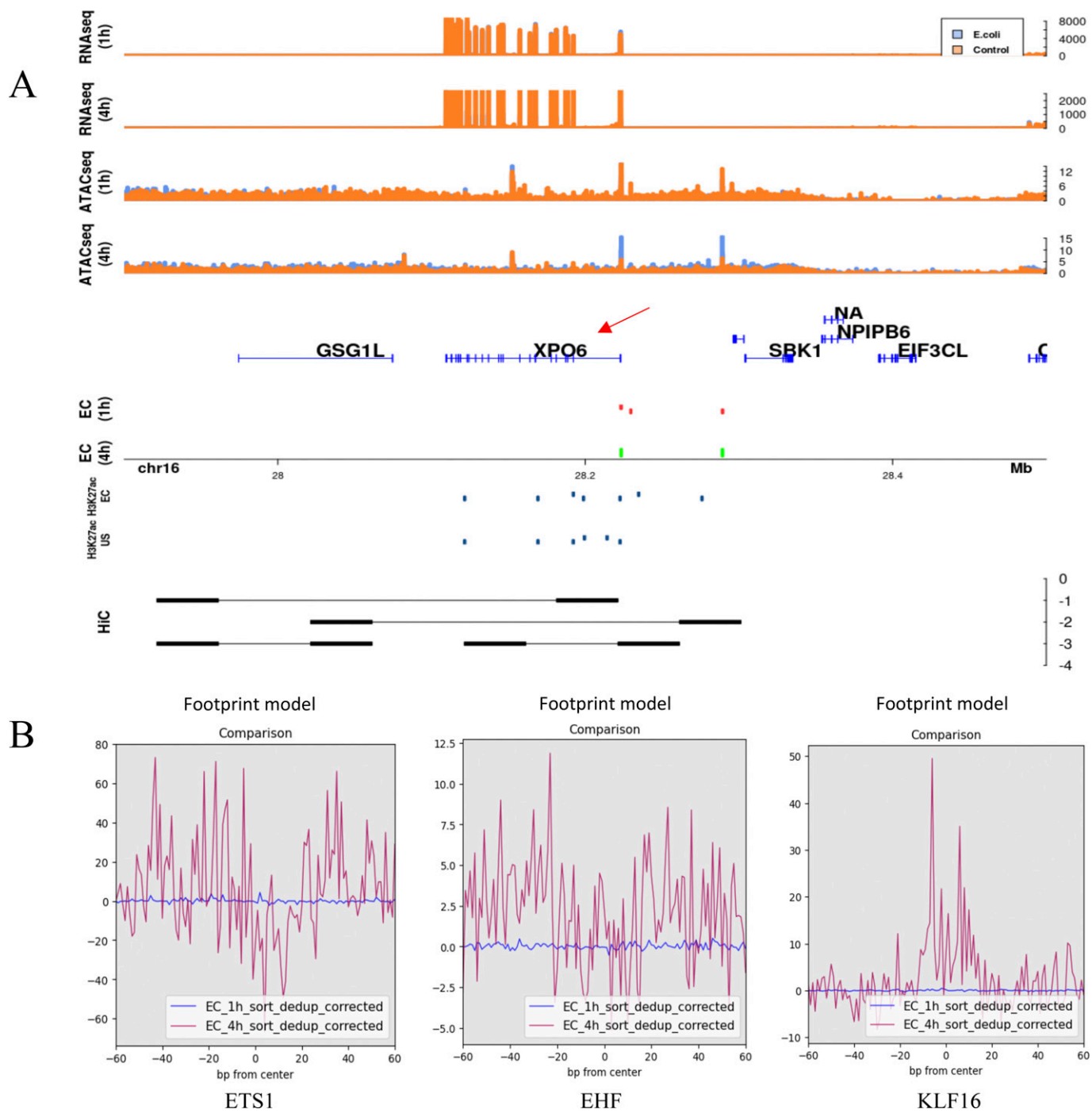

**Figure 6. Intricate combinatorial chromatin accessibility regulation of transcription in neutrophils.**
**(A)** Third mechanism of gene regulation in neutrophils involves differential regions in both the promoter as well as distal sites. An example in this category is the *XPO6* gene. The differential region in the promoter is fixed and unique to an *Escherichia coli* infection. Tracks are similar to those in Fig 5. Differentially closed regions are shown in red. **(B)** Red arrow points to the gene of interest (B). Transcription factor footprinting of enriched motifs identified in the distal associated differential regions. Footprinting using the ATAC-seq reads was performed using TOBIAS (63). Time-dependent binding of transcription factors affecting gene expression.

Similarly, at 4 h of *E. coli* challenge, processes involved in neutrophil activation and degranulation, response to molecules of bacterial origin, intrinsic apoptotic signaling pathway, SRP-dependent co-translational protein targeting to membrane, and processes using autophagy mechanisms are enriched in the up-regulated genes.

More diverse processes are enriched in the down-regulated genes at 4 h and interestingly, many overlaps with those enriched in the up-regulated genes at 4 h. As expected, however, many processes involved in immune response are enriched in the genes that are up-regulated at both time points. These include neutrophil migration,

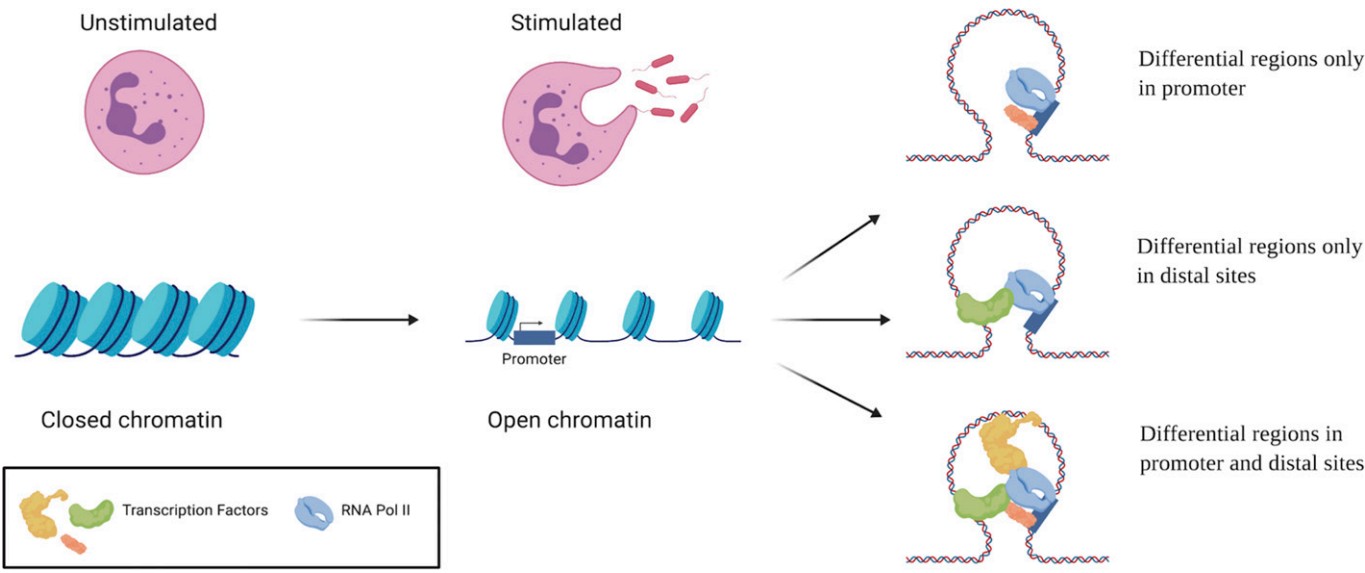

**Figure 7. Model for intricate accessible chromatin regulation of gene expression in neutrophils.**
A schematic depicting the three mechanisms of regulation identified in our analyses. Upon exposure to stimuli, the closed chromatin in neutrophils opens up in stimulus specific patterns. Accessible chromatin regulation of genes occurs in one of three mechanisms where (1) differential regions occur only in the promoter region; (2) differential regions occur only at distal cites; and (3) differential regions occur at promoter and distal sites.

cellular response to lipopolysaccharides, positive regulation of inflammatory response, and chemokine-mediated signaling. Surprisingly there are no enriched processes in the continuously down-regulated genes (Fig S4).

## Transcriptional plasticity of neutrophils is a result of complex accessible chromatin crosstalk

Combination of ATAC-seq and RNA-seq revealed a complex combination of differentially open and closed chromatin regions affecting gene expression changes. For each category of gene expression, a count of the whether the associated peaks are differentially closed or open at each time shows that there is no specific pattern (Fig 4B). That is, genes being up-regulated at 1 h are not all a result of just differential chromatin regions either opening or closing. For genes that are differentially expressed only at 1 h, there likely are chromatin accessibility changes occurring at 4 h that are maintaining gene expression level with the control. This is readily apparent when looking at the distribution of the openness or closedness of associated differential regions (Fig 4B). This phenomenon is clear when surveying the associated differential regions for each gene (Fig 4C). For example, of the 16 genes that are down-regulated at 1 h and up-regulated at 4 h, differential chromatin regions were found to be associated only with two genes, COX16 and uracil phosphoribosyltransferase (UPRT). Each of these genes has two associated DRs that affect gene expression. For COX16, two associated DRs are induced at 1 h, whereas for UPRT, 1 associated DR is induced at 1 and 4 h. Interestingly, however, there is no direct correlation between gene expression and the number of associated differential regions (Spearman correlation–EC1h: 0.0321 EC4h: 0.0289). In general, there are more chromatin accessibility changes earlier, whereas transcriptional changes largely occur at the later time point.

Based on the in silico–linked chromatin accessibility changes and the transcriptional expression, the regulatory mechanisms were classified into three proposed categories: (1) differential regions only in the promoter (1 h: 899; 4 h: 553); (2) differential regions only in distal sites, whereas the promoter region is primed for expression (1 h: 4,675; 4 h: 2,181); and (3) differential regions in the promoter and distal sites (1 h: 633; 4 h: 90). Although these three mechanisms are prevalent, there are differentially expressed genes that do not have any associated differential chromatin regions. In the first category, differential regions were present only in the TSS ± 2.5 kb regions (Fig 5A). An example for this category is the PLSCR1 gene which encodes phospholipid scramblase 1 and is related to the EGF/EGFR signaling pathway. This repressed DR in the promoter region results in ~3 logFC increase in gene expression at 4 h in the presence of E. coli. In the second category, regions in the promoter are open; however, there is no difference between the E. coli challenge in comparison to control. In this category, gene expression is fine-tuned by distal regulatory regions. For example, the TNFAIP3 gene, encoding the tumor necrosis factor α induced protein 3, is up-regulated at both time points (~3 logFC at 1 h and ~7 logFC at 4 h) and has two associated induced distal DRs at 1 h and one induced distal DR at 4 h. These regions being open facilitate the maintenance of gene expression at both time points (Fig 5B). The third category is a combination of differential chromatin regions in the promoter region and in associated distal sites. An example is the XPO6 gene that encodes exportin 6 and is a member of the importin-β family. It is down-regulated ~ −1.3 logFC only at 4 h (Fig 6A). Interestingly, the accessible chromatin signature associated is complex. A differential chromatin region is fixed in the promoter at both time points, repressed at 1 h and induced at 4 h, and yet differential gene expression occurs only at one. This is likely a result of an interaction between the promoter DR and the associated distal regions—two that are repressed at 1 h and one induced at 4 h. These distal interactions are further supported by the Hi-C predicted interacting regions (19). These distal regions

fall within 226,569 Hi-C–predicted interactions across the genome. A complex interplay of interacting chromatin regions facilitates expression of the associated gene. These interactions facilitate the activation or repression of binding motifs to fine tuning the regulation. Transcription factor footprinting of enriched motifs in open distal differential regions associated with the *XPO6* gene revealed a temporal pattern of binding (Fig 6B) similar to the patterns of differentially accessible chromatin regions. The motifs ETS1, EHF, and KLF16 are bound at 4 h but not at 1 h. A combination of varied differentially accessible chromatin regions and transcription factor binding motifs provide an intricate means of transcriptional regulation in neutrophils.

# Discussion

In this study, we explore the chromatin accessibility signatures affecting immune response in neutrophils challenged with external infecting stimuli using ATAC-seq. ATAC-seq on neutrophils provides a unique advantage, interrogating the host response biology, while simultaneously gathering information about pathogens with higher detection sensitivity. Identifying the pathogen responsible for sepsis is an area of intense research. In addition, ATAC-seq can evaluate the accessibility changes in intergenic and enhancer regions (31), showing cell type specific enhancer usages. Even though next-generation sequencing has provided technological advances, gathering useful host and microbial information simultaneously is challenging. Only a fraction of the reads generated from human samples correspond to pathogens and, even if successfully retained in the sea of human DNA, contig assembly is very difficult, so classification must be able to proceed accurately with short reads from noisy data (32, 33). There is a large body of work dedicated to defining positivity via setting thresholds (adjusted based on pathogen load and misclassification due to incomplete reads), species identification to differentiate pathogen from contaminant, and subtraction from negative controls (32, 33). Our assay design improves sensitivity by capitalizing on negative isolation of neutrophils, reducing the amount of background human DNA and cell-free microbial DNA in the final sequencing sample and then only sequencing open chromatin from the eukaryote and prokaryote. It performs drastically better than standardized diagnostic library preparation methods (Fig 1E). This assay requires a small sample volume making it ideal towards clinical adoption in sepsis diagnosis.

Neutrophils are known to form subpopulations at the site of inflammation in response to the stimulus (34), but it was unclear whether epigenetic changes are challenge specific. We find challenge specific genome wide chromatin changes (Fig 2B–D) despite similar genomic distributions (Fig 2A). Although no NETs were believed to be formed as a result of the stimulations, future work is required to accurately determine the exact time chromatin decompensation begins in response to stimuli. Nonetheless, the unique chromatin changes observed reflect in diverse enriched pathways as well specific transcription factor binding motifs with respect to each challenge (Fig 3). There are no common enriched pathways across all challenges, which supports the challenge-specific response nature. Reflective of the ~120 differential regions common to the two *E. coli* challenge time points, their associated enriched pathways are varied (Fig 3A). Interestingly, however, there are not many shared enriched pathways

between *E. coli* and the corresponding ligand, LPS portraying the differences between single ligand and whole organism stimulation. These chromatin accessibility signatures support the earlier discovered stimulus specific gene expression changes in response to LPS and *E. coli* (35) and are unlikely to be artifacts of different stimulation strategies. The unique chromatin accessibility signatures also expose unique transcription factor binding motifs specific to certain challenges (Fig 3B and Table S2). These early and unique chromatin accessibility signatures and exposure of transcription factor binding motifs potentially lead to distinctive downstream responses, namely NETs (36), in response to different stimuli.

Early events in immune cells involved in sepsis can and should be captured in their epigenome, as this is the first step in the cellular response to a cell's environment. These chromatin accessibility events could be a source for new diagnostic tools and even novel molecular targets for new therapies. At 1 h, under many stimuli in this first responder cell type, we find challenge specific genome wide changes in chromatin accessibility (Fig 2B–D). This phenomenon is also readily apparent when comparing the differential chromatin accessibility at the two time points of the challenges (5,010 at 1 h versus 1,688 at 4 h). Measurements of gene and protein expression capture events much later than epigenomic changes and hence may be less informative. For example, it has been shown that enhancer profiling was better at determining cell identity than mRNA (37). This delayed transcription versus epigenetics is supported by our data. Whereas more chromatin accessibility changes are observed at 1 h, more differential expression of genes occurred at 4 h and in a time specific manner (Figs 4B and C and S4). Hence, we propose that a combination of unique differentially accessible chromatin regions and motif signatures we have identified may be more illuminating of the neutrophil's pathogen exposure. These exposure-specific chromatin accessibility changes are rapidly induced and, whereas many maybe transient, may leave a longer lasting "mark" on the epigenome, potentially spawning a new forensic and diagnostic modality, an advantage over current tools, as the epigenome is the earliest detectable signal.

Plasticity in neutrophils has been widely accepted recently (38, 39), but the mechanisms driving this have yet to be successfully delineated. Current focus on the role of epigenetics has vastly expanded the understanding (40), but much is still unknown. A study of unchallenged neutrophils from healthy volunteers identified more than 2,000 genes with a significant epigenetic component explaining their expression (41) and the role of epigenetics in sepsis induced immunosuppression in various immune cells has been identified (38). Subsequently, significant chromatin restructuring was observed in response to a 3 h *E. coli* infection (19). Although these chromatin restructures facilitate the opening of the inflammatory response armament, the exact accessible chromatin interactions are still unknown. With the *E. coli* infection time series in our study, we find that both transcription and chromatin accessibility are plastic in neutrophils. This is readily apparent in the low overlap in the differential regions between the two time points. This can be attributed to the changes in chromatin structure and the need for opening/closing of transcription factor binding sites giving rise to transient gene expression. Based on the differential regions and gene associations, we classified three broad categories of accessible chromatin regulation that occur within earlier identified CTCF-anchored loops (19). The categories include differential regions only in the promoter,

differential regions only in the distal enhancer regions, and finally genes regulated by a combination of both. Of these categories, genes with the differential regions only within the promoter were the fewest (Fig 5A) and showed new H3K27ac modifications within the promoter. Genes with primed promoter regions being regulated only by distal enhancer regions were the highest (Fig 5B) and showed histone marks in the promoter, suggesting promoter activation, under both challenged and unstimulated states further supporting our classification. The third category is a combination of both differential regions in promoters as well as distal regions and the *XPO6* gene, is an example of how this results in gene expression (Fig 6A). Although similar mechanisms of epigenetic transcriptional regulation are known in macrophages (42, 43) and the additive or competitive roles of multiple distal enhancers for gene are known (44, 45), this is the first evidence for such regulation in neutrophils. We also see time specific binding of transcription factors (Fig 6B) within the XPO6 associated distal regions, possibly involved in cooperative activation or repression similar to other systems (46, 47, 48, 49) or by inhibiting the binding of different transcription factors (20) to result in the observed gene expression. Overall, we show that neutrophils undergo plastic transcriptional expression under intricate accessible chromatin regulation that is unique to the stimulus faced (Fig 7) and that this methodology can potentially be used in combination with these signatures as a putative diagnostic tool.

# Materials and Methods

### Study participants

Four healthy volunteer females 30–40 yr old were recruited and informed consent obtained (Stanford University Institutional Review Board-37618).

### Negative selection isolation and activation of neutrophils

#### FACs and flow cytometric analysis
Neutrophils were isolated using a negative selection method that allowed for seamless dovetailing with the ATAC-seq method (Figs 1A and S5). Neutrophils were isolated using the Stem Cell EasySep Direct Human Neutrophil Isolation Kit as per the manufacturer's protocol and resuspended in PBS with 1% FBS and 2 mM EDTA. $8.0 \times 10^5$ cells per condition were fixed in 1% PFA for 10 min at room temperature and subsequently stained with primary antibodies: mouse anti-human PEcy7-CD16 (BD #557744), mouse anti-human PerCPcy5.5-CD66b (#305107; BioLegend), and mouse anti-human V500-CD45 (#560779; BD) as well as the corresponding IgG controls. Flow cytometric and statistical analysis were performed using FlowJo V. 10.0.8.

### TLR samples

Neutrophils isolated from four healthy volunteers were plated at 50,000 cells per well and stimulated in duplicate with the following ligands for 1 h: lipotechoic acid (LTA) 100 ng/ml (Invivogen), LPS (Sigma-Aldrich) 100 ng/ml, Flagellin (FLAG) 300 ng/ml, resiquimod

(R848) (Invivogen), 10 $\mu M$, CpG Class C ODN 2395 5 $\mu M$, BGP 100 $\mu g/ml$ (Invivogen), and HMGB1 (R&D) 1 $\mu g/ml$ (4, 5, 25, 26) (Fig 1B).

### Live organism challenge samples

Blood from two healthy volunteers was spiked with a specific CFU/ml of either *E. coli* American Type Culture Collection 25922 or *S. aureus* American Type Culture Collection 29312. The Stem Cell EasySep Direct Human Neutrophil Isolation Kit was applied as per the manufacturer's protocol to 2 ml of blood after 1 h of *S. aureus* treatment and 1 and 4 h of *E. coli* treatment. After isolation, cells were counted, divided into 50,000 cell samples (50 $\mu l$) in duplicate (Fig 1B).

### Quantitative RT-PCR of IL8 and TNFα

Total RNA was isolated from $8.0 \times 10^5$ cells prepared as above with the RNeasy kit (QIAGEN). RNA was DNase treated using the TURBO DNA-free DNase treatment (Ambion). One step qRT-PCR was performed in the Rotor Gene Q using the Rotor Gene SYBR Green RT-PCR kit. ΔΔCt was calculated using GAPDH. Primer sets are as follows: *IL8 F* 5′CAGTTTTGCCAAGGAGTGCT, *IL8 R* 5′ACTTCTCCACAACCCTCTGC, *TNF F* 5′GCTGCACTTTGGAGTGATCG, *TNF R* 5′ATGAGGTACAGGCCCTCTGA, *GAPDH F* 5′TGCACCACCAACTGCTTAGC, *GAPDH R* 5′GGCATGGACTGTGGTCATGA.

### SYTOX assays for neutrophil extracellular traps

Neutrophils were isolated as above using either the TLR sample or live organism sample preparation as appropriate. Cells were plated at $2.0 \times 10^5$ per well in triplicate. A positive control was created by stimulating cells with 25 nM PMA (Sigma-Aldrich). 5 mM SYTOX green (Life Technologies) was used to detect the presence of NETs (50). Fluorescence intensity was measured using the Tecan Infinite M200 Pro.

### ATAC-seq and RNA-seq library preparation and sequencing

All treated and untreated control cells from four donors and six ligands as well as two donors and two whole organism challenges were collected as described above and ATAC-seq was performed as described (31). Excess primers libraries were removed using the AMPpure bead kit. In parallel with ATAC-seq, genome-wide sequencing using a standard SPRI (51) library preparation was performed on the *S. aureus* challenged neutrophils with AMPure XP from Beckman Coulter.

RNA was extracted from isolated neutrophils after the 1- and 4-h *E. coli* challenges as well untreated controls using the miRNeasy Micro kit from QIAGEN and libraries were generated using KAPA PolyA enrichment mRNA library prep. All libraries were sequenced at the Stanford Functional Genomics Facility on the Illumina HiSeq.

### ATAC-seq analysis

#### Data processing and peak calling
Fastq files were analyzed from raw data all the way to peak calls using the PEPATAC pipeline (http://pepatac.databio.org/en/latest/) against the hg19 build of the human genome. Briefly, reads were trimmed of adapters using Trimmomatic (52) and aligned to hg19 using Bowtie2 (53) with the very-sensitive -X 2000 parameters.

Duplicates were removed using PICARD tools (http://broadinstitute.github.io/picard/). Reads with Mapping quality (MAPQ) <10 were filtered out using SAMtools (54). Reads mapping to the mitochondria or chromosome Y were removed and not considered. Technical replicates were merged using SAMtools yielding one sample per donor per stimulation. Peaks were called using MACS2 (55) with the -q 0.01 −shift 0−nomodel parameters. Correlation between replicates was generated and a single peakset was generated across replicates for each challenge.

## Microbial identification

Reads generated from both SPRI as well as ATAC-seq were pre-processed by trimming with Trimmomatic. Using Kraken (56), human reads were removed from the samples and relative abundances for pathogens were determined as counts per million reads. Replicates were averaged together and $\log_2$ transformed for an abundance value.

## Differential analysis

Differentially accessible regions were identified from the merged peak sets using the DiffBind R package (57). A $P$-value of 0.05 and abs(logFC)≥1 were set as the threshold. Consensus bed files were generated with Diffbind with a threshold of 0.66 overlap. Overlapping/common regions between peak sets were determined using the DiffBind tool and visualized with the UpSet package (58) in R. Induced and repressed differential regions in response to specific challenges were defined with respect to their corresponding controls.

## Assigning genes associated with differential regions

Differential regions in each sample were associated with genes following multiple approaches: (1) T-gene (28 *Preprint*) from the MEME suite was used to predict regulatory links between the differential region and the genes. Only associations with $P$-value < 0.05 and correlation ≥ 0.4 were included. (2) Using GREAT (29) and implementing the basal plus extension algorithm and defining a 2.5 kb region each for proximal upstream and downstream, respectively, and a distal region up to 500 kb. (3) Surveying overlap of differential regions with previously reported association links (30, 59) using BEDTools (60). Additional support for these associations was derived by incorporating predicted Hi-C interactions from *E. coli* stimulation of neutrophils for 3 h (19). A functional pathway enrichment analysis was performed using the ChIPseeker (61) package on R with the custom developed table with the differential regions and their associated genes. This uses a hypergeometric model to assess the enrichment of genes associated with a pathway. A Benjamin-Hochberg adjusted $P$-value of 0.01 was used as a cutoff.

## Annotating differential regions

Genomic distribution of differential regions and enrichment around TSSs were estimated using ChIPseeker. A custom background of histone marks was collected from the International Human Epigenome Consortium (http://ihec-epigenomes.org). We

selected for mature neutrophil samples and women of Northern European ancestry. Sample files in bigBed format were converted to bed format using the UCSCtools package. Motif analysis was performed using hypergeometric optimization of motif enrichment (HOMER) (62) and known motifs with a cutoff of $P < 10^{-10}$ were selected to choose for enriched motifs. Each differential region was annotated with known overlapping histone marks and a list of motifs. Presence−absence heat maps of the enriched motifs in each sample were plotted using heatmap.2 from within the gplots package in R. Footprinting of transcription factors enriched in regions of interest was performed using TOBIAS (63).

## RNA-seq analysis

### *Data processing and differential expression analysis*
Quality of the paired-end reads generated for each replicate was performed using FastQC (64) and trimmed with Trim Galore (https://github.com/FelixKrueger/TrimGalore). Resulting reads were aligned to hg19 using HISAT2 (65) with the−rna-strandness RF parameter. The generated SAM files were sorted and then converted to BAM using SAMtools. Counts were generated using the R package Rsubread (66) in a strand specific manner. Differential gene expression analysis was performed using edgeR (67) and genes with FDR corrected $P$-value < 0.05 and logFC ≥ 1 or logFC ≤ −1 were selected. GO term enrichment analysis was performed and comparisons between time points were made using the compareCluster function from the cluterProfiler R package (68) which uses a hypergeometric model to assess the enrichment of genes associated with a pathway. A Benjamin−Hochberg adjusted $P$-value of 0.01 was used as a cutoff.

# Data Availability

The RNA-seq and ATAC-seq data generated in this study are deposited in the Gene Expression Omnibus under accession numbers GSE153521 and GSE153520, respectively. The data can also be found as a University of California Santa Cruz (UCSC) genome browser session at https://genome.ucsc.edu/s/nikhilram/hg19_LSA_submission. All the codes used to generate data and plots in this study are available at https://github.com/nikhilram/neutrophil_ATACseq.

# Supplementary Information

# Acknowledgements

We would like to thank the Stanford Functional Genomics Facility for performing the sequencing carried out in this study. All bioinformatics analyses were performed on the Stanford Genomic Cluster of the Stanford Research Computing Center. We would also like to thank Shin Lin MD, PhD, Assistant Professor of Cardiology at University of Washington, for critical review and feedback of the manuscript. HY Chang is an Investigator of the Howard

Hughes Medical Institute. SA Thair was supported by a Post Doctoral Research Fellowship from the Stanford Child Health Research Institute. HY Chang is supported by NIH P50-HG007735 and the Scleroderma Research Foundation.

## Author Contributions

N Ram-Mohan: data curation, formal analysis, methodology, and writing—original draft, review, and editing.
SA Thair: data curation, formal analysis, methodology, and writing—review and editing.
UM Litzenburger: methodology and writing—review and editing.
S Cogill: formal analysis and writing—review and editing.
N Andini: methodology and writing—review and editing.
X Yang: methodology and writing—review and editing.
HY Chang: conceptualization, funding acquisition, and writing—review and editing.
S Yang: conceptualization, supervision, funding acquisition, project administration, and writing—original draft, review, and editing.

## Conflict of Interest Statement

HY Chang is a co-founder of Accent Therapeutics, Boundless Bio, and an advisor to 10× Genomics, Arsenal Biosciences, and Spring Discovery. No other authors have competing interests to report.

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
