## [Reviewer comments · Life Science Alliance]

Life Science Alliance

Profiling chromatin accessibility responses in human neutrophils with sensitive pathogen detection.

Nikhil Ram Mohan, Simone Thair, Ulrike Litzenburger, Steven Cogill, Nadya Andini, Xi Yang, Howard Chang, and Samuel Yang

DOI: <https://doi.org/10.26508/lsa.202000976>

Corresponding author(s): Samuel Yang, Stanford University and Samuel Yang, Stanford University

Review Timeline:

Submission Date:	2020-12-02
Editorial Decision:	2021-01-06
Revision Received:	2021-03-02
Editorial Decision:	2021-05-25
Revision Received:	2021-06-03
Editorial Decision:	2021-06-04
Revision Received:	2021-06-07
Accepted:	2021-06-08

Transaction Report:

January 6, 2021

Re: Life Science Alliance manuscript #LSA-2020-00976-T

Dr. Samuel Yang
Stanford University
950 Welch Road
Suite 350
Stanford, CA 94305

Dear Dr. Yang,

Thank you for submitting your manuscript entitled "Profiling chromatin accessibility responses in human neutrophils with sensitive pathogen detection." to Life Science Alliance. The manuscript was assessed by expert reviewers, whose comments are appended to this letter.

We would like to invite you to submit a revised version of this manuscript that addresses the reviewers' points. As you will note from the reviewer comments, the reviewers found the findings intriguing but have also pointed out some technical concerns and requests for various controls raised by the reviewers, (Rev #1 and Rev #3), which should be addressed in the revised manuscript. Follow-up functional experiments and GRO-seq analysis of E. coli - stimulated neutrophils will not be required for further consideration at LSA.

Thank you for this interesting contribution to Life Science Alliance. We are looking forward to receiving your revised manuscript.

Sincerely,

Shachi Bhatt, Ph.D.
Executive Editor
Life Science Alliance
<https://www.lsjournal.org/>
Tweet @SciBhatt @LSAJournal

- A letter addressing the reviewers' comments point by point.
- An editable version of the final text (.DOC or .DOCX) is needed for copyediting (no PDFs).
- High-resolution figure, supplementary figure and video files uploaded as individual files: See our detailed guidelines for preparing your production-ready images, <https://www.life-science-alliance.org/authors>
- Summary blurb (enter in submission system): A short text summarizing in a single sentence the study (max. 200 characters including spaces). This text is used in conjunction with the titles of papers, hence should be informative and complementary to the title and running title. It should describe the context and significance of the findings for a general readership; it should be written in the present tense and refer to the work in the third person. Author names should not be mentioned.

B. MANUSCRIPT ORGANIZATION AND FORMATTING:

Reviewer #1 (Comments to the Authors (Required)):

This manuscript by Ram-Mohan and colleagues describes a study investigating the in vitro effect of a variety of different external stimuli on the chromatin structure and gene expression profile of human neutrophils. The authors have enriched neutrophils from four healthy volunteers, have challenged these cells with an array of different ligands or with whole E. coli or S. aureus bacteria in vitro, and then have conducted ATACseq and RNAseq to assess changes in chromatin accessibility and in the transcriptome. The authors report the following main results: 1. ATACseq of

neutrophils can facilitate detection of pathogen DNA in host cells and as such may improve pathogen recognition in sepsis patients. 2. Pathogen challenging of neutrophils results in ligand-specific chromatin structure reorganization (opening of promoters and putative distal enhancer elements). 3. These chromatin alterations do only in part explain ligand-induced changes in gene expression over time.

This is a descriptive but informative study. Its main strength is that the authors have compared the impact of a great variety of different ligands and of two different bacteria species (stimulating different host cells' recognition receptors) on the chromatin accessibility landscape of neutrophils. However, the major weakness is that none of their findings was followed after with functional experiments, so that evaluation of biological significance is lacking. Therefore, the reader is left with uncertainty on the relevance of these findings. Based on the lack of functional validation, it is unclear why the different challenges lead to different open chromatin landscapes. Do these chromatin differences reflect any known differences in how neutrophils react to the various pathogens or ligands? Moreover, the paper mentions sepsis several times, but there is no direct linkage between the data and sepsis in this study.

In detail, I have the following critique points:

Regarding results chapter "Neutrophils are activated in response to ..." and associated Fig. 1: The authors should measure apoptosis/cell death in their in vitro cultured neutrophils in the presence and absence of ligands. Moreover, they should verify in greater depth that the neutrophils' chromatin stayed intact during culture and stimulation. Just looking at NET formation is not sufficient. Western blotting against histone H3 according to Branzk et al. 2014, Nature Immunology 15, p. 1017-1025, which is more sensitive in reporting chromatin decompensation than NET formation would be better. The authors should also show at least three biological replicates with all different challenges throughout this study. Finally, the authors should show untreated controls in Fig. 1C.

Regarding results chapter "Pathogen DNA from challenges ...". The authors performed negative isolation of neutrophils only for ATAC-seq but not for the SPRI library preparation. Equal starting cell material would make both techniques more comparable, considering sensitivity for pathogen DNA. The way the authors have performed this experiments leaves uncertainty as to whether the enriched neutrophils or the ATAC approach facilitated pathogen DNA detection.

Regarding results chapter "Differential accessibility of chromatin in the genome ..." and associated Fig. 2 and 3 as well as tables: The authors state that they have used a combination of different prediction tools for annotating ATAC peaks to gene pathways. How were the results prioritized in cases where different tools predicted different ATAC peak to gene associations? How were the predictions validated? Moreover, how useful are these annotations at all? Not all open chromatin regions are functional enhancer elements. In addition, assigning an ATAC peak to a certain gene does not necessarily mean that this gene is expressed in the neutrophils or is regulated by the different challenges. Finally, how do the annotated gene sets and pathways explain differences in how neutrophils react to them? Fig. 2A: Again, the authors need to show untreated controls. Moreover, are the differences in ATAC peak distributions between the samples significant? Fig. 2B: Numbers at the y-axis are unreadable. The legend (lines 785-786) states TCF7L2 gene, next sentence TLE3 gene. In the figure, TLE3 is shown. Which one is it? Fig. 3B and C: These figures are too small and should be better explained in the legends. Moreover, unstimulated controls are missing. I would also like to see motif names in the figures. Tables 1 and 2: I suggest combining both tables, showing an overview on how many DARs (differential accessible regions) were identified per stimuli compared to the unstimulated control in total + how many of these were unique for the

respective stimulus. This way, it would become much clearer that in all cases over 60 % of accessible chromatin, regions are challenge specific.

Regarding results chapter "Transcriptional plasticity of neutrophils is a result of complex ..." and associated Fig. 4: The authors find that differences in chromatin accessibilities are not well linked with transcriptomic changes in *E. coli*-stimulated neutrophils. One reason for this lack of linkage could be that with RNAseq the authors did not measure de novo gene expression. Hence, they should perform global run-on sequencing (GROseq) to detect newly synthesized transcripts, which should be a much better readout for their question. I also would like to see names of differentially expressed example genes mentioned in the text. Fig. 4A: The authors need to show reproducibility between the biological replicates. Fig. 4B: As most genes are not differentially expressed (NC, grey), the smaller bars (containing the more interesting information) are hard to see. Maybe change axes.

Reviewer #2 (Comments to the Authors (Required)):

In this manuscript the authors examine how chromatin accessibility in neutrophils is altered upon exposure to an ensemble of ligands. The data show distinctive changes in chromatin accessibility, cis-tomic elements and transcription signatures. The authors also demonstrate how chromatin accessibility is altered in a time-dependent manner. Based on patterns of nucleosome depletion the authors identify three categories of transcriptional regulation in neutrophils: (i) chromatin accessibility changes in promoter regions, (ii) alterations in the promoter and distal enhancers and (iii) changes in gene expression associated with distal enhancers.

This is an interesting paper. The assay developed here improves sensitivity by reducing the non-accessible human DNA as well as cell free microbial DNA in the final sequencing sample. The sensitivity is further enhanced by sequencing accessible chromatin derived from both neutrophils and microbial agents.

Comment:

The authors compare their data to recently published HiC reads for chromosome 6. However, paper could be further strengthened by comparing patterns of chromatin folding observed in recent HiC studies across the entire genome for microbial exposed human neutrophils. Do the ATAC-Seq sensitive regions map to genomic regions associated with changes in remote genomic interactions when analyzed at a genome-wide scale? Could the authors please include that analysis in their manuscript?

Reviewer #3 (Comments to the Authors (Required)):

Ram-Mohan and colleagues used a combined approach of ATAC-seq and RNA-seq to profile the response of human neutrophils in bacterial infection. First, the authors highlight the possibility of using ATAC-seq instead of whole genome sequencing to enrich for the bacterial DNA as diagnostic tool in sepsis. Further, the authors analysed the changes in the chromatin states in response to a set of TLR ligands describing a highly specific epigenetic regulation of neutrophil activation. The manuscript represent a valuable piece in the understanding of neutrophil biology, a cell type long-thought to be not able to regulate transcription and almost not capable of producing new proteins once reaching the mature state. Nevertheless, the manuscript also presents some criticalities as

follows:

Major points:

1. In Figure 1D, the authors show the NETosis formation for the isolated TLR ligands. It would be of interest to see also the NETosis levels of SA- and EC-stimulated neutrophils at 1 hour and for EC at 4h since those are conditions analysed later in the manuscript.
2. The authors explain the mapping of reads to other bacterial strains as short reads not mapping specifically. It is not clear to the reviewer why this effect is so specific for the samples infected with SA and is not found in the untreated samples.
3. Figure 2B looks really strange, probably for some formatting problems with the pdf. It is difficult to interpret the results here.
4. In Figure 2C and 2D the authors show a dramatically specific chromatin remodelling in response to the different stimuli. Nevertheless, this analysis relies on a p-value and a FC cut-off. It would be of interest to visualize, if - for example - the genes found to be specific for one condition are also regulated in a similar direction in other conditions (e.g., with a FC/FC plot).
5. Figure 5A: it appears that the background level in the 4h EC-stimulated neutrophils is much higher, it would be important to know the viability and the NETosis activity of those cells at the moment of the analysis to have a clear interpretation of the results (see point 1).
6. This reviewer finds it difficult to interpret the comparison between the LPS-stimulated neutrophils and the EC-treated blood. In fact, the protocol of those two stimulations is quite different. In the first case, only the purified neutrophils were exposed to LPS where in the second the stimulation was performed in whole blood and the neutrophils were isolated later. This aspect should be carefully discussed in the manuscript and the statement comparing the two stimulations should also be carefully revised in light of the different protocols.
7. A code availability statement is missing: considering the highly computational nature of the manuscript the authors need to deposit the code used for the analysis in public repositories such as GitHub. Similarly, the data should be provided in the respective databases. Even better, data, analysis and code could be provided as a package on platforms that allow such integrated view on the analysis performed. E.g., see FASTGenomics.org as an example for such a transparent and integrated representation of what was done.

Minor points:

1. Line 133, 134: The reviewer understands that the TLR ligands used are well known in the field but referring to some publications there would help the reader to delve the topic.
2. Figure 1C and 1D, a dashed line at $y = 1$ would help the reader to better understand the difference of the treated cells compared to control.
3. Figure 1B, the order of the stimuli is not consistent with the other plots.
4. Figure S1 is only mentioned in the text of the methods section, if not mentioned before Figure S2 the authors should consider re-numbering. Furthermore, there is a mislabel of the CD16-CD66b dotplot since both axes are named CD16.
5. Figure 1E: the label WB (whole blood) is a little bit confusing since the sequencing should be of the negative-selected neutrophils.
6. Line 198-200: This is a really interesting statement the author could provide a figure to visualize it.
7. It is not completely clear: what is the final number of genes mapped from the differentially open regions (line 204-223)? The authors need to provide this information in the text and as column in the table of the differential open regions (Table S1).
8. Figure 5A, 5B and 6A: the authors need to guide the reader more in the figures indicating the

genomic regions mentioned in the text.

9. Line 453: alpha is in bold.

We would like to convey our sincere gratitude to the editor and the reviewers for a complete and thorough critique of our work and all the suggestions that have made our manuscript better. We have addressed all of the reviewers' comments below individually and either made edits to the manuscript as per their suggestions or justified our choice. Short of re-running some of the suggested functional experiments, we have modified the manuscript to accommodate most of the reviewers' suggestions. Please find our responses to individual comments below -

Reviewer #1 (Comments to the Authors (Required)):

This manuscript by Ram-Mohan and colleagues describes a study investigating the in vitro effect of a variety of different external stimuli on the chromatin structure and gene expression profile of human neutrophils. The authors have enriched neutrophils from four healthy volunteers, have challenged these cells with an array of different ligands or with whole *E. coli* or *S. aureus* bacteria in vitro, and then have conducted ATACseq and RNAseq to assess changes in chromatin accessibility and in the transcriptome. The authors report the following main results: 1. ATACseq of neutrophils can facilitate detection of pathogen DNA in host cells and as such may improve pathogen recognition in sepsis patients. 2. Pathogen challenging of neutrophils results in ligand-specific chromatin structure reorganization (opening of promoters and putative distal enhancer elements). 3. These chromatin alterations do only in part explain ligand-induced changes in gene expression over time.

This is a descriptive but informative study. Its main strength is that the authors have compared the impact of a great variety of different ligands and of two different bacteria species (stimulating different host cells' recognition receptors) on the chromatin accessibility landscape of neutrophils. However, the major weakness is that none of their findings was followed after with functional experiments, so that evaluation of biological significance is lacking. Therefore, the reader is left with uncertainty on the relevance of these findings. Based on the lack of functional validation, it is unclear why the different challenges lead to different open chromatin landscapes. Do these chromatin differences reflect any known differences in how neutrophils react to the various pathogens or ligands? Moreover, the paper mentions sepsis several times, but there is no direct linkage between the data and sepsis in this study.

Thank you for your in-depth critique of our manuscript, we appreciate all the feedback. The goal of this manuscript was to demonstrate that the responses of neutrophils to stimuli is not restricted to the transcriptional level but expands to the epigenomic level as well. In fact, we show that these epigenomic changes are challenge specific. We believe that exploring the functional aspect of these epigenomic changes is beyond the scope of this manuscript. Neutrophils are known to produce Neutrophil Extracellular Traps of heterogenous protein compositions in response to different stimuli (Petretto et al., 2019) and now our study shows that these differences are likely a result of early epigenomic changes.

A statement reflecting the above is now included in the discussion in lines 415 - 417 and reads – ‘These early and unique chromatin accessibility signatures and exposure of transcription factor binding motifs potentially lead to distinctive downstream responses, namely NETs (Petretto et al., 2019), in response to different stimuli.’

Although we did not directly study septic patients, the challenges we tested are representative of common pathogens that cause sepsis. We believe that our findings provide evidence for how epigenomic changes occur in these immune cells in response to systemic infections caused by bacterial or viral pathogens.

In detail, I have the following critique points:

Regarding results chapter "Neutrophils are activated in response to ..." and associated Fig. 1: The authors should measure apoptosis/cell death in their in vitro cultured neutrophils in the presence and absence of ligands. Moreover, they should verify in greater depth that the neutrophils' chromatin stayed intact during culture and stimulation. Just looking at NET formation is not sufficient. Western blotting against histone H3 according to Branzk et al. 2014, Nature Immunology 15, p. 1017-1025, which is more sensitive in reporting chromatin decompensation than NET formation would be better. The authors should also show at least three biological replicates with all different challenges throughout this study. Finally, the authors should show untreated controls in Fig. 1C.

Thank you for the interesting comment. We did not culture neutrophils in vitro, all the challenges were performed on either whole blood or fresh neutrophils extracted from donor blood. Although it is a great suggestion to perform Western blotting against the histone H3, we do not believe it is a feasible option for us to perform. Given that we performed all of our experiments on fresh neutrophils isolated from donor blood, we cannot retroactively perform the Western blotting. Additionally, since we only needed to ensure overall intactness of the chromatin for ATAC-seq, looking for NETs was a sufficient indicator to proceed.

We chose to perform the analyses on merged peaksets for each challenge since the overall correlation between replicates was high (r^2 between replicates for each challenge was between 0.7 and 0.9). Additionally, while generating merged peaksets, we only considered peaks in replicates that overlapped, no peaks unique to replicates were included.

Figure 1C represents the fold change in the expression of IL8 and TNF in challenges with respect to the untreated controls, that is why we do not include the untreated controls in the figure.

Regarding results chapter "Pathogen DNA from challenges ...". The authors performed negative isolation of neutrophils only for ATAC-seq but not for the SPRI library preparation. Equal starting cell material would make both techniques more comparable, considering sensitivity for pathogen DNA. The way the authors have performed this experiments leaves uncertainty as to whether the enriched neutrophils or the ATAC approach facilitated pathogen DNA detection.

Thank you for pointing this out. We did, in fact, perform SPRI and ATAC-seq on negatively isolated neutrophils. We apologize for the lack of clarity in the text and have now edited the main text and figure legend to reflect the above –

Main text on lines 152 – 153: ‘To demonstrate this, whole blood was challenged with SA in incremental colony forming units (CFU) per mL for 1 hour and neutrophils were negatively isolated’.

Figure 1 legend (lines 783 - 787): ‘Whole blood was spiked at increasing concentrations with live SA and neutrophils were negatively isolated. These were prepared for sequencing using traditional DNA extraction and library preparation (SPRI method) compared to neutrophil isolation and ATAC-seq method. (n = 2) (Neu: isolated neutrophils, no organisms).’

Regarding results chapter "Differential accessibility of chromatin in the genome ..." and associated Fig. 2 and 3 as well as tables: The authors state that they have used a combination of different prediction tools for annotating ATAC peaks to gene pathways. How were the results prioritized in cases where different tools predicted different ATAC peak to gene associations? How were the predictions validated? Moreover, how useful are these annotations at all? Not all open chromatin regions are functional enhancer elements. In addition, assigning an ATAC peak to a certain gene does not necessarily mean that this gene is expressed in the neutrophils or is regulated by the different challenges. Finally, how do the annotated gene sets and pathways explain differences in how neutrophils react to them? Fig. 2A: Again, the authors need to show untreated controls. Moreover, are the differences in ATAC peak distributions between the samples significant? Fig. 2B: Numbers at the y-axis are unreadable. The legend (lines 785-786) states TCF7L2 gene, next sentence TLE3 gene. In the figure, TLE3 is shown. Which one is it? Fig. 3B and C: These figures are too small and should be better explained in the legends. Moreover, unstimulated controls are missing. I would also like to see motif names in the figures. Tables 1 and 2: I suggest combining both tables, showing an overview on how many DARs (differential accessible regions) were identified per stimuli compared to the unstimulated control in total + how many of these were unique for the respective stimulus. This way, it would become much clearer that in all cases over 60 % of accessible chromatin, regions are challenge specific.

Thank you for these comments. Below are our responses to each of the reviewer's comments –

The authors state that they have used a combination of different prediction tools for annotating ATAC peaks to gene pathways. How were the results prioritized in cases where different tools predicted different ATAC peak to gene associations? How were the predictions validated?

The peak to gene associations by the different tools used were not prioritized. In an effort to be comprehensive in our peak-gene associations and to reduce any artificial bias, we included all peak-gene associations called by the different tools used. The peak-gene associations that were obtained from the ENCODE overlap were previously validated

associations and the calls from T-Gene and GREAT were only filtered out for stringent p-values. All associated genes for each differential peak are listed in supplemental table 1.

Moreover, how useful are these annotations at all? Not all open chromatin regions are functional enhancer elements. In addition, assigning an ATAC peak to a certain gene does not necessarily mean that this gene is expressed in the neutrophils or is regulated by the different challenges. Finally, how do the annotated gene sets and pathways explain differences in how neutrophils react to them?

These are interesting points and we thank the reviewer for making them. These annotations provide us with a simple framework to describe what these differentially accessible chromatin regions might be doing, determining the function of which is beyond the scope of the study. It is indeed true that assigning a differentially accessible chromatin region to a gene does not necessarily mean that the gene is expressed in response to the different challenges. However, given that we see multiple patterns in the number of associated genes, it gives us a basis to hypothesize the intricate regulatory mechanisms that affect gene regulation. Additionally, as the reviewer notes, relying on gene expression alone might not give us enough distinctive patterns to help differentiate the response of the neutrophils to different stimuli and drives the need to look for these differentially accessible chromatin regions in the first place. Figure 3A describes how these differences in the peak-gene association across challenges are represented in the functional pathways. Here, we do not describe pathways that are actually enriched in transcription rather the enrichment of differentially accessible regions in genes in these pathways.

Fig. 2A: Again, the authors need to show untreated controls. Moreover, are the differences in ATAC peak distributions between the samples significant?

Figure 2 represents only differentially accessible regions and not the raw peaks themselves. Since the differentially accessible regions are determined with respect to the untreated controls, we believe that representing that peaks for the untreated control would result in the figure describing 2 different things. Additionally, since we are interested in the specific regions that are differentially accessible across the challenges rather than the number of differentially accessible regions between the challenges, we did not look for significance in the numbers of differentially accessible regions.

Fig. 2B: Numbers at the y-axis are unreadable. The legend (lines 785-786) states TCF7L2 gene, next sentence TLE3 gene. In the figure, TLE3 is shown. Which one is it? Fig. 3B and C: These figures are too small and should be better explained in the legends. Moreover, unstimulated controls are missing. I would also like to see motif names in the figures. Tables 1 and 2: I suggest combining both tables, showing an overview on how many DARs (differential accessible regions) were identified per stimuli compared to the unstimulated control in total + how many of these were unique for the respective stimulus. This way, it would become much clearer that in all cases over 60 % of accessible chromatin, regions are challenge specific.

Thank you for pointing out the error and the many useful suggestions. Figure 2B is now fixed and the tables have been combined as per the reviewer's suggestion. With figures 3B and 3C, since enriched motifs were determined only in the differentially accessible regions, we do not include the untreated controls. We would prefer to not show the motif names on

the heatmaps in figures 3B and 3C for readability. Below is the heatmap for the enriched motifs in induced regions for an example. The figure would be very busy with information that is already present in supplemental tables 2 and 3. Supplemental table 2 lists the unique motifs and supplemental table 3 lists all the motifs enriched in the induced and repressed differential regions. Additionally, we intended for figure 3B to be a presence/absence plot of the information present in the supplemental material which further shows that responses to stimuli are unique even at the enriched motif level.

[Figure removed by editorial staff per authors' request].

Regarding results chapter "Transcriptional plasticity of neutrophils is a result of complex ..." and associated Fig. 4: The authors find that differences in chromatin accessibilities are not well linked with transcriptomic changes in E. coli-stimulated neutrophils. One reason for this lack of linkage could be that with RNAseq the authors did not measure de novo gene expression. Hence, they should perform global run-on sequencing (GROseq) to detect newly synthesized transcripts, which should be a much better readout for their question. I

also would like to see names of differentially expressed example genes mentioned in the text. Fig. 4A: The authors need to show reproducibility between the biological replicates. Fig. 4B: As most genes are not differentially expressed (NC, grey), the smaller bars (containing the more interesting information) are hard to see. Maybe change axes.

Thank you for the suggestion. However, since the focus of the manuscript is primarily the epigenomic changes in neutrophils in response to different challenges, we do not think that additional functional experiments are within the scope of this study. It would be an interesting follow up study. The y axis for figure 4B is now represented in a log10 scale as requested. Strong correlation is observed in the counts for each gene between the replicates for RNA-seq as well (r^2 ranging from 0.92 to 0.99). We have now included a statement about the same on line 248.

Reviewer #2 (Comments to the Authors (Required)):

In this manuscript the authors examine how chromatin accessibility in neutrophils is altered upon exposure to an ensemble of ligands. The data show distinctive changes in chromatin accessibility, cis-tromic elements and transcription signatures. The authors also demonstrate how chromatin accessibility is altered in a time-dependent manner. Based on patterns of nucleosome depletion the authors identify three categories of transcriptional regulation in neutrophils: (i) chromatin accessibility changes in promoter regions, (ii) alterations in the promoter and distal enhancers and (iii) changes in gene expression associated with distal enhancers.

This is an interesting paper. The assay developed here improves sensitivity by reducing the non-accessible human DNA as well as cell free microbial DNA in the final sequencing sample. The sensitivity is further enhanced by sequencing accessible chromatin derived from both neutrophils and microbial agents.

Comment:

The authors compare their data to recently published HiC reads for chromosome 6. However, paper could be further strengthened by comparing patterns of chromatin folding observed in recent HiC studies across the entire genome for microbial exposed human neutrophils. Do the ATAC-Seq sensitive regions map to genomic regions associated with changes in remote genomic interactions when analyzed at a genome-wide scale? Could the authors please include that analysis in their manuscript?

Thank you for your comment. We did survey the overlap between the differentially accessible regions we observe after stimulation with *E. coli* for 1 and 4 hours and the Hi-C interactions observed after an *E. coli* exposure of 3 hours (Denholtz et al., 2020).

For 3 of the examples we listed in figures 5 and 6, we represent the total number of interacting regions predicted by the above study in context of the examples we are listing and describe the differentially accessible regions we find. As a genome wide analysis, we looked for the overlap of our differentially accessible regions and the Hi-C predicted

interacting regions. The genome wide overlap results are represented in the main text in lines 202 to 208 and reads – ‘Additionally, comparing EC1h and EC4h differential regions with Hi-C defined interacting regions after 3 hours of treatment with *E. coli* show overlap between the ATAC-seq differential regions and Hi-C predicted interacting regions. 4,506 EC1h differential regions overlap with one end of the Hi-C interactions while 4,498 overlap with the other. Similarly, for EC4h, 1,494 differential regions overlap with one end of the interacting regions and 1,471 overlap with the other.’

Reviewer #3 (Comments to the Authors (Required)):

Ram-Mohan and colleagues used a combined approach of ATAC-seq and RNA-seq to profile the response of human neutrophils in bacterial infection. First, the authors highlight the possibility of using ATAC-seq instead of whole genome sequencing to enrich for the bacterial DNA as diagnostic tool in sepsis. Further, the authors analysed the changes in the chromatin states in response to a set of TLR ligands describing a highly specific epigenetic regulation of neutrophil activation. The manuscript represent a valuable piece in the understanding of neutrophil biology, a cell type long-thought to be not able to regulate transcription and almost not capable of producing new proteins once reaching the mature state. Nevertheless, the manuscript also presents some criticalities as follows:

Major points:

1. In Figure 1D, the authors show the NETosis formation for the isolated TLR ligands. It would be of interest to see also the NETosis levels of SA- and EC-stimulated neutrophils at 1 hour and for EC at 4h since those are conditions analysed later in the manuscript.

Thank you for this insightful suggestion. We have now included the sytox green assay for the detection of NETs in response to the whole ligand challenges in figure 1 as per the reviewer’s suggestion. Similar to the results of the assay at 1 hour post stimulation with EC, there is no evidence for NETosis even at 5 hours.

Text on line 142 now reads No NETs were observed in response to any stimuli at the time of ATAC-seq (1 or 4 hours of stimulation, Fig 1D).

2. The authors explain the mapping of reads to other bacterial strains as short reads not mapping specifically. It is not clear to the reviewer why this effect is so specific for the samples infected with SA and is not found in the untreated samples.

This is an interesting point. We process all the reads to filter out any that map to the human genome before they are assigned to a species. Reads from untreated samples map very well to the human genome and hence do not result in enough conspicuous ones that result in incorrect classification. However, with the SA infected samples, once the human reads were filtered out, the remaining short reads have an increased chance of being misclassified due to non-specific mapping, possibly to closely related to bacterial genomes.

The results section has been edited to reflect the preprocessing of reads to remove ones that map to the human genome. Lines 158-160 now read – ‘Contaminant signals are

present, after removing all human reads, given that the neutrophils were only challenged with SA, these are likely short, low complexity reads that do not map specifically'.

3. Figure 2B looks really strange, probably for some formatting problems with the pdf. It is difficult to interpret the results here.

Thank you for highlighting the issue with the figure. This is now fixed. We have included a better representation of the differences in differentially accessible regions around the TLE3 gene.

4. In Figure 2C and 2D the authors show a dramatically specific chromatin remodelling in response to the different stimuli. Nevertheless, this analysis relies on a p-value and a FC cut-off. It would be of interest to visualize, if - for example - the genes found to be specific for one condition are also regulated in a similar direction in other conditions (e.g., with a FC/FC plot).

Figures 2C and 2D represent the overlap in differentially accessible chromatin regions across the challenges, and not the differential expression of genes, we apologize for any misunderstanding. We believe that the direction/whether the region either opens or closes in response to a challenge does not add much to differentiating between the challenges since the specific regions themselves are largely unique. Our hypothesis is that different challenges result in epigenomic modifications in specific locations in the neutrophil's chromatin.

5. Figure 5A: it appears that the background level in the 4h EC-stimulated neutrophils is much higher, it would be important to know the viability and the NETosis activity of those cells at the moment of the analysis to have a clear interpretation of the results (see point 1).

Thank you for pointing this out, we have now included the sytox green assay for the detection of NETs in response to the whole ligand challenges in figure 1 as per the reviewer's suggestion. Similar to the results of the assay at 1 hour post stimulation with EC, there is no evidence for NETosis even at 5 hours. The increased background at 4h seems to be a result of an inherent temporal increase in transcription in neutrophils.

6. This reviewer finds it difficult to interpret the comparison between the LPS-stimulated neutrophils and the EC-treated blood. In fact, the protocol of those two stimulations is quite different. In the first case, only the purified neutrophils were exposed to LPS where in the second the stimulation was performed in whole blood and the neutrophils were isolated later. This aspect should be carefully discussed in the manuscript and the statement comparing the two stimulations should also be carefully revised in light of the different protocols.

Thank you for pointing out this important distinction. We have modified the text to reflect the different stimulation strategies.

Lines 196-199 now read: 'There are 3 common differential regions between the SA and LTA challenges, 118 common between the two EC time points, EC1h and LPS have 19 in common, and EC4h and LPS have 3 in common despite the different stimulation strategies (neutrophil vs whole blood)'

Lines 367-369: 'These chromatin accessibility signatures support the earlier discovered stimulus specific gene expression changes in response to LPS and EC (Zhang et al., 2004) and are unlikely to be artifacts of different stimulation strategies'

Since our findings show that epigenomic changes are stimulus specific similar to the gene expression changes observed earlier (Zhang et al., 2004) even using same stimulation strategies, we believe that these unique responses can be attributed to actual responses to challenges rather than the different stimulation strategies we employed.

7. A code availability statement is missing: considering the highly computational nature of the manuscript the authors need to deposit the code used for the analysis in public repositories such as GitHub. Similarly, the data should be provided in the respective databases. Even better, data, analysis and code could be provided as a package on platforms that allow such integrated view on the analysis performed. E.g., see FASTGenomics.org as an example for such a transparent and integrated representation of what was done.

All data generated, both RNA-seq and ATAC-seq, in this study are available in the Gene Expression Omnibus under GSE153521 and GSE153520 and are listed on lines 563 and 564. We have also generated a UCSC Genome Browser session for your perusal –

https://genome.ucsc.edu/s/nikhilram/hg19_LSA_submission

All the code used to generate data and plots in this study are now available at https://github.com/nikhilram/neutrophil_ATACseq.

Minor points:

1. Line 133, 134: The reviewer understands that the TLR ligands used are well known in the field but referring to some publications there would help the reader to delve the topic.

We have now added citations for the TLR ligands we tested.

2. Figure 1C and 1D, a dashed line at $y = 1$ would help the reader to better understand the difference of the treated cells compared to control.

Thank you for the suggestion. We have now added dashed lines in both panels in the figure.

3. Figure 1B, the order of the stimuli is not consistent with the other plots.

Thank you for pointing this out. We have reordered the list of stimuli in Figure 1B to match the order of the challenges in the other plots.

4. Figure S1 is only mentioned in the text of the methods section, if not mentioned before Figure S2 the authors should consider re-numbering. Furthermore, there is a mislabel of the CD16-CD66b dotplot since both axes are named CD16.

Thank you for pointing this out. We have reordered the supplemental figures. The new supplemental figure S5 is also modified to have the axes labeled correctly in the CD16-CD66b dotplot.

5. Figure 1E: the label WB (whole blood) is a little bit confusing since the sequencing should be of the negative-selected neutrophils.

WB is now changed to Neu to represent neutrophils.

6. Line 198-200: This is a really interesting statement the author could provide a figure to visualize it.

Thank you for the suggestion. Since we elude to the overlap between the differential regions detected using ATAC-seq with the Hi-C interacting regions in figures 5 and 6, we chose not to include another figure for the whole genome level analysis. Additionally, given that the ends of the Hi-C interactions could overlap with ones from other interactions, that is, a region that is one end of an interacting pair could be the other end of a different interacting pair, a global representation might be too complex.

7. It is not completely clear: what is the final number of genes mapped from the differentially open regions (line 204-223)? The authors need to provide this information in the text and as column in the table of the differential open regions (Table S1).

Thank you for the suggestion. We have now included a column in Table 1 with the exact number of differential regions for each challenge that were associated with genes by our methods. The associated genes are also listed in supplemental table 1.

8. Figure 5A, 5B and 6A: the authors need to guide the reader more in the figures indicating the genomic regions mentioned in the text.

Thank you for the suggestion, the figures now include red arrows that point to the gene of interest and figure legends are modified accordingly to suggest the same.

9. Line 453: alpha is in bold.

This is now fixed.

References:

- Denholtz, M., Zhu, Y., He, Z., Lu, H., Isoda, T., Döhrmann, S., Nizet, V., & Murre, C. (2020). Upon microbial challenge, human neutrophils undergo rapid changes in nuclear architecture and chromatin folding to orchestrate an immediate inflammatory gene program. *Genes & Development*, *34*(3–4), 149–165. <https://doi.org/10.1101/gad.333708.119>
- Petretto, A., Bruschi, M., Pratesi, F., Croia, C., Candiano, G., Ghiggeri, G., & Migliorini, P. (2019). Neutrophil extracellular traps (NET) induced by different stimuli: A comparative proteomic analysis. *PLOS ONE*, *14*(7), e0218946. <https://doi.org/10.1371/journal.pone.0218946>
- Zhang, X., Kluger, Y., Nakayama, Y., Poddar, R., Whitney, C., DeTora, A., Weissman, S. M., & Newburger, P. E. (2004). Gene expression in mature neutrophils: Early responses to inflammatory stimuli. *Journal of Leukocyte Biology*, *75*(2), 358–372. <https://doi.org/10.1189/jlb.0903412>

May 25, 2021

Re: Life Science Alliance manuscript #LSA-2020-00976-TR

Dr. Samuel Yang
Stanford University
950 Welch Road
Suite 350
Stanford, CA 94305

Dear Dr. Yang,

Thank you for submitting your revised manuscript entitled "Profiling chromatin accessibility responses in human neutrophils with sensitive pathogen detection." to Life Science Alliance. The manuscript has been seen by the original reviewers whose comments are appended below.

We apologize for this extended and unusual delay in getting back to you. As you will see from the comments below, Reviewers 2 and 3 are satisfied with the revised manuscript and ask only for some minor edits. Reviewer 1, however, is not convinced that the identified chromatin changes are likely the cause of differences in NET formation.

Our general policy is that papers are considered through only one revision cycle; however, given the positive outlook and enthusiasm from 2 reviewers, we would like to give you a chance to send us a revised manuscript that includes Histone H3 western blotting data, as requested by Reviewer 1.

Please submit the final revision within one month, along with a letter that includes a point by point response to the remaining reviewer comments. Please let us know if you need more time.

- A letter addressing the reviewers' comments point by point.
- An editable version of the final text (.DOC or .DOCX) is needed for copyediting (no PDFs).
- High-resolution figure, supplementary figure and video files uploaded as individual files: See our detailed guidelines for preparing your production-ready images, <https://www.life-science-alliance.org/authors>
- Summary blurb (enter in submission system): A short text summarizing in a single sentence the study (max. 200 characters including spaces). This text is used in conjunction with the titles of

papers, hence should be informative and complementary to the title and running title. It should describe the context and significance of the findings for a general readership; it should be written in the present tense and refer to the work in the third person. Author names should not be mentioned.

B. MANUSCRIPT ORGANIZATION AND FORMATTING:

Sincerely,

Shachi Bhatt, Ph.D.
Executive Editor
Life Science Alliance
<http://www.lsajournal.org>
Tweet @SciBhatt @LSAJournal

Reviewer #1 (Comments to the Authors (Required)):

The authors have done not much to address my critique points other that trying to discuss them away. For example, my concern on the unclear biological relevance of their finding that different external stimuli cause differences in open chromatin has been address by the authors by the wild speculation that the identified chromatin changes are likely the cause for differences in NET formation. They have shown nothing to provide evidence for such a statement. Moreover, I expressed concern that simply looking at NET formation is to crude as a readout for structurally intact chromatin. Chromatin could already begin to dissolve under in vitro conditions before NETs are seen, so rather histone H3 western blotting as a more sensitive method should be performed. However, the authors argued that they could not retroactively perform such a western blot on the neutrophils they had already used for their chromatin studies. However, this is in my eyes a poor excuse as they simply could isolate and in vitro stimulate new neutrophils for such a Western blot in order to demonstrate that their experimental protocol leaves the neutrophils' chromatin fully intact. It is obviously your editorial decision how to proceed, but my recommendation is against acceptance of this manuscript in its current state.

I have no further comments on this manuscript.

Reviewer #2 (Comments to the Authors (Required)):

This study has profiled chromatin accessibility in human neutrophils in response to pathogens. The authors have addressed my comments. It is now ready for publication.

Reviewer #3 (Comments to the Authors (Required)):

Ram-Mohan and colleagues analyse the changes in the chromatin states of human neutrophils in

response to both isolated TLRs agonists or whole bacteria challenge with *S. aureus* and *E. coli*. The authors also highlight ATAC-seq as a candidate method for an accurate diagnosis of bacterial infection in sepsis. The authors appropriately responded to the concern raised in my review. Nevertheless, there are still few minor points that the author should address before this review can support publication of the manuscript.

Minor Points:

1. The upset plot in figure 2C and D do not seem to reflect totally the description in the text. For example, "1.582 regions were common between a combination of two challenges" (line186) seems a high number compared to what seen in figure 2C where summing the interaction shared by two comparisons would be much smaller. Here also the 0 of the axes is not visible in the plot.
2. Line 270 refers to Fig S5 in the context of gene set enrichment analysis but Fig S5 is the gating strategy. The author probably wanted to refer to figure S4. Nevertheless, the terms named in lines 265-270 are not reflecting the figure, for example "SRP-dependent co-translational protein targeting to membrane" is only up-regulated at 4h but not down-regulated at 1, and the terms listed in line 268-267 are not present.

Reviewer #1 (Comments to the Authors (Required)):

The authors have done not much to address my critique points other than trying to discuss them away. For example, my concern on the unclear biological relevance of their finding that different external stimuli cause differences in open chromatin has been addressed by the authors by the wild speculation that the identified chromatin changes are likely the cause for differences in NET formation. They have shown nothing to provide evidence for such a statement. Moreover, I expressed concern that simply looking at NET formation is too crude as a readout for structurally intact chromatin. Chromatin could already begin to dissolve under in vitro conditions before NETs are seen, so rather histone H3 western blotting as a more sensitive method should be performed. However, the authors argued that they could not retroactively perform such a western blot on the neutrophils they had already used for their chromatin studies. However, this is in my eyes a poor excuse as they simply could isolate and in vitro stimulate new neutrophils for such a Western blot in order to demonstrate that their experimental protocol leaves the neutrophils' chromatin fully intact. It is obviously your editorial decision how to proceed, but my recommendation is against acceptance of this manuscript in its current state.

I have no further comments on this manuscript.

We completely agree that the histone H3 western blotting would be a more sensitive test. The NET assay is a rather crude way to look for intact nuclei and as the reviewer mentions, we do not know if any chromatin decompensation begins before NET formation in our stimulations. We have included a statement in the second paragraph of the discussion on lines 357 -360 that reads –

“Although no NETs were believed to be formed as a result of the stimulations, future work is required to accurately determine the exact time chromatin decompensation begins in response to various stimuli”

Although we did not perform the histone H3 western blotting, we believe our data are representative of changes in chromatin in response to various stimuli since in addition to the lack of NETs using the sytox assay, we have also generated high quality ATAC-seq data which is dependent on the intactness of the nuclei being processed.

Regarding the reviewer's concern about the speculation that the observed differences in epigenomic changes in response to stimuli could result in the variations in NETs formed, we do not claim that these directly result in differences in NET formation. Given that neutrophils have been shown to form diverse NETs in response to different stimuli and we show in this article that unique epigenomic responses are triggered in response to different

stimuli, we merely hypothesize in the discussion a probable connection between the two in the following words –

“These early and unique chromatin accessibility signatures and exposure of transcription factor binding motifs potentially lead to distinctive downstream responses, namely NETs [36], in response to different stimuli.”

Reviewer #2 (Comments to the Authors (Required)):

This study has profiled chromatin accessibility in human neutrophils in response to pathogens. The authors have addressed my comments. It is now ready for publication.

Thank you very much for your in-depth critique of our manuscript.

Reviewer #3 (Comments to the Authors (Required)):

Ram-Mohan and colleagues analyse the changes in the chromatin states of human neutrophils in response to both isolated TLRs agonists or whole bacteria challenge with *S. aureus* and *E. coli*. The authors also highlight ATAC-seq as a candidate method for an accurate diagnosis of bacterial infection in sepsis. The authors appropriately responded to the concern raised in my review. Nevertheless, there are still few minor points that the author should address before this review can support publication of the manuscript.

Thank you very much for your in-depth critique of our manuscript that helped improve the article.

Minor Points:

1. The upset plot in figure 2C and D do not seem to reflect totally the description in the text. For example, "1.582 regions were common between a combination of two challenges" (line186) seems a high number compared to what seen in figure 2C where summing the interaction shared by two comparisons would be much smaller. Here also the 0 of the axes is not visible in the plot.

Thank you for this comment. The upset plots represent the top 100 intersections and since the number of intersections between any pairs is much fewer than the unique peaks, the bar plots for these look much smaller. The figure legend has now been edited on lines 776 and 779 to include the ‘top 100’ overlapping differential regions across ligand challenges and whole organism and corresponding ligand challenges respectively.

There are indeed a total of 1,582 regions common between a combination of any two challenges. Below is the breakdown –

	BGP	R848	FLAG	HMGB	LTA	LPS	EC1h	EC4h	SA
BGP		56	43	35	34	58	29	4	11
R848			264	123	38	64	25	36	10
FLAG				139	55	70	47	7	12
HMGB					31	38	37	6	5
LTA						80	11	2	3
LPS							19	3	5
EC1h								118	45
EC4h									19
SA									

Since our goal was to highlight the fact that even though there were all these shared differential regions, the vast majority of the differential regions were unique to each challenge, we focused on showcasing the number of unique peaks and do not include the pairwise breakdown.

Thank you for noting the missing 0 on the axes. This is now fixed.

2. Line 270 refers to Fig S5 in the context of gene set enrichment analysis but Fig S5 is the gating strategy. The author probably wanted to refer to figure S4. Nevertheless, the terms named in lines 265-270 are not reflecting the figure, for example "SRP-dependent co-translational protein targeting to membrane" is only up-regulated at 4h but not down-regulated at 1, and the terms listed in line 268-267 are not present.

We do refer to figure S4 there, thank you for pointing that out.

Figure S4 is a representation of the top 15 enriched GOterms in each pattern of gene expression we observed in our dataset. Some of the terms in the text that were not represented in the figure were ones that did not fall within the top 15 categories of that gene expression pattern. We have now edited the paragraph to remove any GOterms that are not represented in figure S4. The figure legend for S4 is also updated to read – "GOterm enrichment analysis of the genes categorized based on their pattern over time. Grouped genes analyzed using the enrichGO feature within clusterprofiler. Top 15 GOterm enrichment categories displayed for each expression pattern."

June 4, 2021

RE: Life Science Alliance Manuscript #LSA-2020-00976-TRR

Dr. Samuel Yang
Stanford University
950 Welch Road
Suite 350
Stanford, CA 94305

Dear Dr. Yang,

Thank you for submitting your revised manuscript entitled "Profiling chromatin accessibility responses in human neutrophils with sensitive pathogen detection.". We would be happy to publish your paper in Life Science Alliance pending final revisions necessary to meet our formatting guidelines.

- please upload your supplementary figures as single files as well
- please add your table legends to the main manuscript text after the main and supplementary figure legends

A. FINAL FILES:

-- Summary blurb (enter in submission system): A short text summarizing in a single sentence the study (max. 200 characters including spaces). This text is used in conjunction with the titles of papers, hence should be informative and complementary to the title. It should describe the context and significance of the findings for a general readership; it should be written in the present tense

and refer to the work in the third person. Author names should not be mentioned.

B. MANUSCRIPT ORGANIZATION AND FORMATTING:

Sincerely,

June 8, 2021

RE: Life Science Alliance Manuscript #LSA-2020-00976-TRRR

Dr. Samuel Yang
Stanford University
950 Welch Road
Suite 350
Stanford, CA 94305

Dear Dr. Yang,

Thank you for submitting your Research Article entitled "Profiling chromatin accessibility responses in human neutrophils with sensitive pathogen detection.". It is a pleasure to let you know that your manuscript is now accepted for publication in Life Science Alliance. Congratulations on this interesting work.

DISTRIBUTION OF MATERIALS:

Again, congratulations on a very nice paper. I hope you found the review process to be constructive and are pleased with how the manuscript was handled editorially. We look forward to future exciting submissions from your lab.

Sincerely,
